# Optimizing Few-Step Generation with Adaptive Matching Distillation

Lichen Bai [* 1]  Zikai Zhou [* 1]  Shitong Shao [1]  Wenliang Zhong [1]  Shuo Yang [2]
Shuo Chen [3]  Bojun Chen [1]  Zeke Xie [1]

## Abstract

Distribution Matching Distillation (DMD) is a powerful acceleration paradigm, yet its stability is often compromised in *Forbidden Zones*—regions where the real teacher provides unreliable guidance while the fake teacher exerts insufficient repulsive force. In this work, we propose a unified optimization framework that reinterprets prior art as implicit strategies to avoid these corrupted regions. Based on this insight, we introduce **Adaptive Matching Distillation (AMD)**, a self-correcting mechanism that utilizes reward proxies to explicitly detect and escape *Forbidden Zones*. AMD dynamically prioritizes corrective gradients via structural signal decomposition and introduces *Repulsive Landscape Sharpening* to enforce steep energy barriers against failure mode collapse. Extensive experiments across image and video generation tasks (e.g., SDXL, Wan2.1) and rigorous benchmarks (e.g., VBench, GenEval) demonstrate that AMD significantly enhances sample fidelity and training robustness. For instance, AMD improves the HPSv2 score on SDXL from **30.64** to **31.25**, outperforming state-of-the-art baselines. These findings validate that explicitly rectifying optimization trajectories within *Forbidden Zones* is essential for pushing the performance ceiling of few-step generative models.

## 1. Introduction

Diffusion models (Song et al., 2020; Karras et al., 2022) have demonstrated remarkable success in visual generation, exhibiting remarkable capabilities in generating diverse, high-resolution images (Dhariwal & Nichol, 2021;

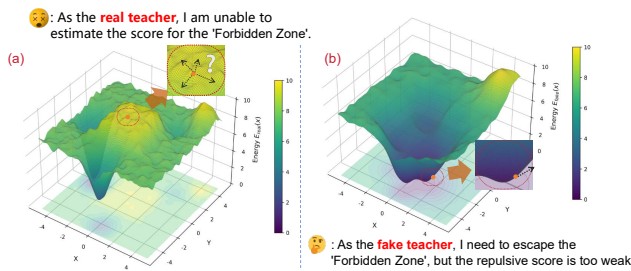

*Figure 1.* Visualization of the Optimization Landscape in *Forbidden Zones*. **(a)** The Real Teacher's energy potential becomes undefined or poorly calibrated far from the data manifold, exerting misleading attractive forces. **(b)** The Fake Teacher's landscape exhibits a shallow slope, resulting in a weak repulsive force that is insufficient to propel the student out of the *Forbidden Zone*.

Podell et al., 2023) and videos (Wan et al., 2025; Wu et al., 2025a; Li et al., 2026; Shao et al., 2026). Despite these advancements, their utility is often hindered by the substantial latency of the iterative denoising process, which demands dozens or even hundreds of Number of Function Evaluations (NFEs) during inference.

To alleviate this latency, a range of distillation methods (Yin et al., 2024a; Salimans & Ho, 2022; Sauer et al., 2024; Liu et al., 2023b; Huang et al., 2025a) have been proposed to compress the multi-step denoising trajectory into a minimal number of generation steps. Among these, Distribution Matching Distillation (DMD) (Yin et al., 2024a) has emerged as a prominent approach. DMD leverages a dual-teacher framework: a real teacher (typically a pre-trained diffusion model) provides gradients to guide the student generated samples toward the target data distribution, while a fake teacher (typically a diffusion model trained alongside the student) approximates the student model's current distribution. Crucially, the fake teacher supplies a repulsive signal to regularize training, encouraging sample diversity and mitigating mode collapse.

Despite its conceptual appeal, the stability of DMD implicitly hinges on a fragile assumption that **the real teacher offers reliable guidance, while the fake teacher provides adequate repulsive signals to guide or push the student samples toward the target distribution** (Shao et al., 2025; Jiang et al., 2025). However, this assumption is frequently compromised in practice. During distillation process, the student model inevitably generates severely distorted or low-

[1]xLeaF Lab, The Hong Kong University of Science and Technology (Guangzhou) [2]Harbin Institute of Technology, Shenzhen [3]School of Intelligence Science and Technology, Nanjing University, China.. Correspondence to: Zeke Xie <zekexie@hkust-gz.edu.cn>.

*Proceedings of the $43^{rd}$ International Conference on Machine Learning*, Seoul, South Korea. PMLR 306, 2026. Copyright 2026 by the author(s).

quality samples that deviate substantially from the real data manifold. For these poor samples, the real teacher's score estimates are often poorly calibrated due to the distribution shift, and the repulsive guidance from the fake teacher proves insufficient to extricate the student from these low-quality regions, which we identify as *Forbidden Zones*. In these zones, as shown in Figure 1, training dynamics may enter a self-reinforcing degradation regime, where the student repeatedly revisits low-quality regions and struggles to recover. Consequently, the lack of effective correction impedes the whole optimization process, hindering convergence and perpetuating generative distortions.

Armed with our definition of *Forbidden Zones*, we offer a novel perspective that unifies various existing DMD methods. We re-interpret these prior works (Yin et al., 2024b;a; Jiang et al., 2025; Shao et al., 2025; Liu et al., 2025a) as methods that implicitly constrain the student to avoid such corrupted regions. Crucially, however, these methods do not explicitly detect the occurrence of *Forbidden Zone*, nor do they provide a direct mechanism to adapt the distillation dynamics once the student inevitably enters one.

To resolve this, we contend that the most effective strategy is to pinpoint these *Forbidden Zone* and execute a swift "leapfrog" back to the valid data manifold. Our framework is founded on three key observations: **(1) Detection via Low Rewards:** Severely degraded samples consistently yield low reward scores. Since these samples lie outside the support of the real teacher, rendering its score estimates unreliable, the reward model serves as a practical proxy for identifying *Forbidden Zones*; **(2) Asymmetric Propulsion:** To facilitate an escape, the fake teacher should not learn the student distribution uniformly; instead, it must specialize in modeling these corrupted regions to provide targeted and potent repulsive forces; and **(3) Signal Prioritization:** Within the *Forbidden Zone*, the repulsive force must take precedence over the potentially misleading guidance from the real teacher to ensure effective correction.

Building on these insights, we propose **AMD** (**A**daptive **M**atching **D**istillation). Unlike previous static distillation approaches, AMD acts as a self-correcting system: it utilizes reward-guided diagnostics to identify distorted samples and adaptively modulates the distillation dynamics, prioritizing specialized signals to actively "push" the student out of the *Forbidden Zone*. The dynamic intervention effectively mitigates the risk of training collapse and significantly promotes higher generation fidelity.

The contributions of this work are threefold:

First, we analyze the *Forbidden Zone* in DMD and reinterpret recent advancements as implicit mechanisms that avoid these corrupted regions. This perspective highlights the fundamental needs to explicitly address *Forbidden Zones* for

stable and effective distillation.

Second, we propose AMD, a reward-aware distillation framework that transforms DMD into a self-correcting system. By leveraging reward models as diagnostic sensors, it adaptively adapts the distillation dynamics: the fake teacher provides targeted repulsive guidance, and real and fake signals are dynamically prioritized to enable rapid recovery from *Forbidden Zones*.

Third, we demonstrate the robust effectiveness of AMD across large-scale image and video generation tasks (e.g., SDXL, Wan2.1). AMD significantly mitigates training collapse while achieving superior sample fidelity. By leveraging reward-aware guidance, AMD enables the student to transcend the performance ceiling of the original teacher, delivering state-of-the-art results on diverse human-preference benchmarks (e.g., VideoGen-Eval, HPSv2).

## 2. Preliminaries

In this section, we present preliminaries about Distribution Matching Distillation in diffusion models. Due to page limitations, a comprehensive discussion of related work is provided in Section B.

**Distribution Matching Distillation**    Distribution Matching Distillation (DMD) (Yin et al., 2024b) compresses a pre-trained multi-step diffusion model (the *real teacher*) into a few-step generator (the *student*) $G_\theta$ by minimizing the KL divergence between the target distribution $p_{\text{real}}$ and the student-induced distribution $p_{\text{fake}}$.

Let $x = G_\theta(z)$ with $z \sim \mathcal{N}(\mathbf{0}, \mathbf{I})$, and let $t \sim \mathcal{U}[0, 1]$ denote a randomly sampled diffusion time. The DMD objective is defined as

$$\mathcal{L}_{\text{DMD}} = -\mathbb{E}_{z,t}\left[\log p_{\text{real}}(\mathcal{F}_t) - \log p_{\text{fake}}(\mathcal{F}_t)\right], \quad (1)$$

where $\mathcal{F}_t(x)$ denotes the forward diffusion operator that injects noise at time $t$, defined as $\mathcal{F}_t(x) = t \cdot x + (1 - t) \cdot \epsilon$ with $\epsilon \sim \mathcal{N}(\mathbf{0}, \mathbf{I})$. Specifically, taking the gradient of Equation (1) with respect to the student parameters $\theta$ yields a compact score-matching form as

$$\nabla_\theta \mathcal{L}_{\text{DMD}} = -\mathbb{E}_{z,t}\left[\left(s_{\text{real}}(\mathcal{F}_t) - s_{\text{fake}}(\mathcal{F}_t)\right)\frac{\partial G_\theta(z)}{\partial \theta}\right]. \quad (2)$$

This formulation manifests a pull-push dynamic centered on the student-generated sample $x = G_\theta(z)$: the real teacher exerts an *attractive score* that guides the student toward the target distribution, whereas the fake teacher applies a *repulsive score* that drives the student away from its own current distribution. This interaction is formally analyzed through an optimization lens in Section 3.1.

**Reward Model** We assume access to a pre-trained reward (or preference) model $\mathcal{R} : \mathcal{X} \to \mathbb{R}$, which assigns a scalar score $\mathcal{R}(x)$ to a generated sample $x \in \mathcal{X}$. The reward model is fixed throughout training and treated as a black-box scoring function.

## 3. Method

In this section, we first re-examine DMD through the lens of optimization, establishing a unified framework to analyze its training dynamics. Based on this formulation, we formally define *Forbidden Zones* and demonstrate how existing methods can be categorized within our framework as implicit mitigation strategies. Finally, we introduce our proposed **AMD**, detailing the theoretical mechanism by which it facilitates a swift return to the valid data manifold.

### 3.1. An Optimization Perspective on DMD

While DMD is conventionally formulated as a distribution matching problem via score estimation, which often obscure the mechanistic causes of training failure. To gain a more actionable understanding, we reformulate the student's parameter-space update as an effective optimization step performed directly on the generated samples. This shift is essential for two reasons: it provides a clear **push-pull** physical intuition of the distillation dynamics, and a mathematical tool for diagnosing training collapse.

**Proposition 3.1** (Sample-space Gradient Descent Reformulation). *(See Appendix C) Let $x = G_\theta(z)$ be a latent sample generated by the student. Under the first-order approximation, the parameter update $\theta \leftarrow \theta - \eta \nabla_\theta \mathcal{L}_{\mathrm{DMD}}$ induces an effective gradient descent step on the sample $x$ in the latent space:*

$$x_{new} = x - \eta_{eff} \cdot \nabla_x \mathcal{V}_{\mathrm{DMD}}(x), \tag{3}$$

*where $\mathcal{V}_{\mathrm{DMD}}(x)$ is the **distillation potential** defined as the contrastive energy between the teacher-guided targets as:*

$$\begin{aligned} \mathcal{V}_{\mathrm{DMD}}(x) &= \frac{1}{2}\|x - \hat{x}_{0,\mathrm{real}}\|^2 - \frac{1}{2}\|x - \hat{x}_{0,\mathrm{fake}}\|^2 \\ &= \frac{1}{2}\underbrace{\|\mathbf{d}_{\mathrm{real}}\|^2}_{\text{Attractive Term}} - \frac{1}{2}\underbrace{\|\mathbf{d}_{\mathrm{fake}}\|^2}_{\text{Repulsive Term}}. \end{aligned} \tag{4}$$

*Taking the gradient of $\mathcal{V}_{\mathrm{DMD}}(x)$ with respect to $x$ yield $\nabla_x \mathcal{V}_{\mathrm{DMD}}(x) = \mathbf{d}_{\mathrm{real}} - \mathbf{d}_{\mathrm{fake}}$. Here $\hat{x}_{0,\mathrm{real}}$ and $\hat{x}_{0,\mathrm{fake}}$ denote the denoised latent estimates from the real and the fake teacher, typically computed via Tweedie's formula from the diffused state $\mathcal{F}_t(x)$.*

From the optimization perspective in Proposition 3.1, the parameter-space update of DMD is equivalent to a push-pull navigation task in the latent space. Here, the student $G_\theta$ reconciles its current position by moving toward the real

teacher's target (attracted by $\mathbf{d}_{\mathrm{real}}$) while moving away from the region estimated by the fake teacher (repelled by $\mathbf{d}_{\mathrm{fake}}$).

Crucially, the reliability of the attractive force $\mathbf{d}_{\mathrm{real}}$ depends on the real teacher's ability to accurately predict clean targets. We define the region where this competence fails as the *Forbidden Zone*:

**Definition 3.2** (Forbidden Zone $\mathcal{Z}_f$). Let $E_{\mathrm{real}}(x) = -\log p_{\mathrm{real}}(x)$ denote the energy potential of the target distribution. The *Forbidden Zone $\mathcal{Z}_f$* is defined as the high-energy regime beyond the teacher's empirical support:

$$\mathcal{Z}_f = \{x \in \mathcal{X} \mid E_{\mathrm{real}}(x) > \gamma\} \tag{5}$$

where $\gamma$ is a threshold of teacher competence. Inside $\mathcal{Z}_f$, the student produces severely distorted samples that lie outside the real teacher's training manifold.

By viewing DMD as a navigator of latent space, we precisely diagnose the systemic optimization failures in $\mathcal{Z}_f$ through the energy landscapes shown in Fig. 1. Specifically, within $\mathcal{Z}_f$, the real teacher's energy surface $E_{\mathrm{real}}$ becomes fractured and ill-posed due to a lack of empirical support (Fig. 1), causing the attractive force $\mathbf{d}_{\mathrm{real}}$ to yield hallucinated and incoherent gradients.

Simultaneously, distorted samples reside at the extreme tails of the student's distribution where the fake teacher's energy $E_{\mathrm{fake}}$ is identically flat (Fig. 1b). This leads to a vanishing repulsive force, depriving the student of the propulsion required to escape the corrupted region. Such a dual collapse results in an optimization stalemate: the noisy pull and negligible push fail to produce a effective force $\nabla_x \mathcal{V}_{\mathrm{DMD}}$, therefore hindering the effectiveness of distillation.

**Re-interpreting Prior Art.** This optimization perspective provides a unified lens for re-interpreting recent advances in the DMD family. Rather than viewing these methods as ad hoc modifications, we express them as specific instantiations of a **generalized DMD gradient operator**:

$$\mathbf{g} = \mathcal{H}\big(\mathbf{d}_{\mathrm{real}}(x, \mathcal{F}_t, \phi), \mathbf{d}_{\mathrm{fake}}(x, \mathcal{F}_t, \psi)\big) + \lambda \cdot \mathbf{\Psi}_{\mathrm{ext}}, \tag{6}$$

where $\mathbf{d} = x - \hat{x}_0$ denotes the latent displacement, and $\mathcal{H}(\cdot, \cdot)$ is a adaptive operator that composes the attractive and repulsive signals induced by the real and fake teachers. Here, $\mathcal{F}_t$ denotes the re-noising scale, and $\mathbf{\Psi}_{\mathrm{ext}}$ represents auxiliary forces beyond the vanilla DMD objective.

Within this framework, substituting Equations (3) and (4) reveals that standard DMD recovers a linear case of $\mathcal{H}$:

$$\mathcal{H}_{\mathrm{std}}(\mathbf{d}_{\mathrm{real}}, \mathbf{d}_{\mathrm{fake}}) = \mathbf{d}_{\mathrm{real}} - \mathbf{d}_{\mathrm{fake}}, \tag{7}$$

which induces a fixed contrastive gradient field and serves as the baseline optimization dynamics. We then demonstrate in Table 1 how Equation (6) unifies various existing methods, with Figure 2 providing visual intuition for this framework.

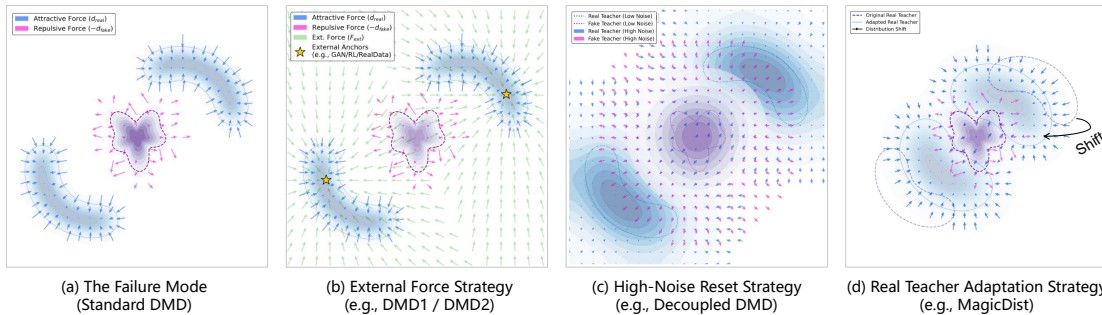

(a) The Failure Mode
(Standard DMD)

(b) External Force Strategy
(e.g., DMD1 / DMD2)

(c) High-Noise Reset Strategy
(e.g., Decoupled DMD)

(d) Real Teacher Adaptation Strategy
(e.g., MagicDist)

*Figure 2.* Visualizing the Taxonomy of Forbidden Zone Mitigation Strategies. **(a)** Standard Failure: Useful gradients exist only near modes, leaving a "gradient vacuum" in $\mathcal{Z}_f$. **(b)** External Force: An auxiliary force $\mathbf{\mathcal{V}}_{\text{ext}}$ (green) provides global steering to bridge the gap. **(c)** Noise Reset: High noise induces distribution overlap, restoring gradient coverage. **(d)** Real Teacher Adaptation: The teacher manifold actively shifts towards the student to eliminate $\mathcal{Z}_f$.

*Table 1.* Taxonomy of DMD-style methods from an optimization perspective. We decompose each method into a configuration of the generalized displacement gradient operator. $\tilde{a}$ denotes the reward-aware advantage, $\phi$ is the real teacher state, and $\psi$ represents the fake teacher optimization objective, where $\mathcal{L}_{\text{diff}} \coloneqq \|\epsilon - \epsilon_\psi(x_t, t)\|^2$.

| Method | DMD | DMD2 | D-DMD | Magic Dist. | DMDR | AMD (Ours) |
|---|---|---|---|---|---|---|
| $\mathcal{H}(\mathbf{d}_{\text{real}}, \mathbf{d}_{\text{fake}})$ | $\mathbf{d}_{\text{real}} - \mathbf{d}_{\text{fake}}$ | $\mathbf{d}_{\text{real}} - \mathbf{d}_{\text{fake}}$ | $\mathbf{d}_{ca} + \mathbf{d}_{dm}$ | $\mathbf{d}_{\text{real}} - \mathbf{d}_{\text{fake}}$ | $\mathbf{d}_{\text{real}} - \mathbf{d}_{\text{fake}}$ | $\alpha\mathbf{d}_{ca} + \beta\mathbf{d}_{dm}$ |
| **Ext. Force** $(\mathbf{\mathcal{V}}_{\text{ext}})$ | $x - x_{gt}$ | $\nabla_x \log D(x)$ | $0$ | $0$ | $\nabla_\theta \mathbb{E}[R(x)]$ | $\mathbf{0}$ |
| **Noise SNR** $(\mathcal{F}_t)$ | $\mathcal{U}[0, 1]$ | $\mathcal{U}[0, 1]$ | $t_\phi > t_G$ | $\mathcal{U}[0, 1]$ | $\mathcal{U}[0, 1]$ | $\mathcal{U}[0, 1]$ |
| **Real Teacher** $(\phi)$ | $\phi_{\text{fix}}$ | $\phi_{\text{fix}}$ | $\phi_{\text{fix}}$ | $\phi_\theta \to \phi_{\text{fix}}$ | $\phi_\theta \to \phi_{\text{fix}}$ | $\phi_{\text{fix}}$ |
| **Fake Teacher** $(\psi)$ | $\mathcal{L}_{\text{diff}}$ | $\mathcal{L}_{\text{diff}}$ | $\mathcal{L}_{\text{diff}}$ | $\mathcal{L}_{\text{diff}}$ | $\mathcal{L}_{\text{diff}}$ | $\mathbf{\mathcal{W}}(\tilde{\mathbf{a}}) \cdot \mathcal{L}_{\text{diff}}$ |
| $\mathcal{Z}_f$ **Strategy** | Static Tethering | Adv. Anchoring | SNR Reset | Manifold Shrink | RL Steering | RL Steering |

Several methods introduce an auxiliary force $\mathbf{\mathcal{V}}_{\text{ext}}$ to provide a non-zero recovery gradient when the primary displacement fields $\mathbf{d}_{\text{real}}$ and $\mathbf{d}_{\text{fake}}$ enter a deadlock. Standard **DMD** (Yin et al., 2024b) utilizes an $L_2$ regression loss, acting as a static tethering force $\mathbf{\mathcal{V}}_{\text{ext}} = x - x_{gt}$ that anchors the student to ground-truth data. **DMD2** (Yin et al., 2024a) replaces this with an adversarial force $\mathbf{\mathcal{V}}_{\text{ext}} = \nabla_x \log D(x)$, creating an adversarial fence to deflect samples away from $\mathcal{Z}_f$. Similarly, **DMDR** (Jiang et al., 2025) alternates between DMD updates and RL steps, where the latter introduces an asynchronous steering force driven by reward gradients to ensure the trajectory remains within high-fidelity regions.

Other methods try to circumvent $\mathcal{Z}_f$ by manipulating either the noise scale $\mathcal{F}_t$ or the teacher state $\phi$. **D-DMD** (Liu et al., 2025a) designs a novel re-noising schedule, its essence lies in lowering the risk of the real teacher estimating ill-posed scores within $\mathcal{Z}_f$ by shifting samples toward more reliable noise regimes. In contrast, **MagicDistillation** (Shao et al., 2025) temporarily pulls the teacher's empirical support toward the student's current distribution ($\phi_\theta \to \phi_{fix}$) via LoRA. This induces a manifold shrinking effect, reducing the measure of $\mathcal{Z}_f$ during early training stages. We provide the corresponding analytical understanding in Section C.2.

### 3.2. Dynamic Score Adaptation via AMD

To address $\mathcal{Z}_f$ pathologies, a straightforward intuition is to rebalance the contribution of the real and fake teachers

based on the sample's current state. Specifically, one might aim to let $\mathbf{d}_{\text{fake}}$ dominate within $\mathcal{Z}_f$ to provide the necessary repulsive "push" for escape, while ensuring $\mathbf{d}_{\text{real}}$ governs high-fidelity regions to refine the distillation. This leads to a **naive adaptive strategy**:

$$\mathcal{H}_{\text{naive}}(\mathbf{d}_{\text{real}}, \mathbf{d}_{\text{fake}}) = \alpha(\tilde{a}) \cdot \mathbf{d}_{\text{real}} - \beta(\tilde{a}) \cdot \mathbf{d}_{\text{fake}}, \quad (8)$$

where $\alpha$ and $\beta$ are scalar coefficients controlled by the reward-aware advantage $\tilde{a}$. However, we contend that such coarse-grained rescaling is fundamentally insufficient.

As analyzed in Section 3.1, the optimization failure within $\mathcal{Z}_f$ stems not merely from insufficient gradient magnitude, but from a structural collapse of the gradient field where teacher guidance becomes ill-posed. As empirically demonstrated in our 2D analysis (Fig. 8 of Section F.1), this can lead to **destructive interference** between attractive and repulsive forces, magnifying conflicting signals that desta­bilize training trajectories and eventually trigger a total dis­tillation collapse. This observation suggests that a more fine-grained, *component-wise* intervention is required to decouple beneficial guidance from detrimental noise.

**Decomposing the Gradient Field.** To resolve this dilemma, we draw inspiration from the analytical decompo­sition proposed in D-DMD (Liu et al., 2025a). Let the real teacher's guidance be $\mathbf{d}_{\text{real}} = \mathbf{d}_{\text{real}}^{\text{uncond}} + \omega(\mathbf{d}_{\text{real}}^{\text{cond}} - \mathbf{d}_{\text{real}}^{\text{uncond}})$, where $\omega$ is the CFG scale. The standard DMD gradient

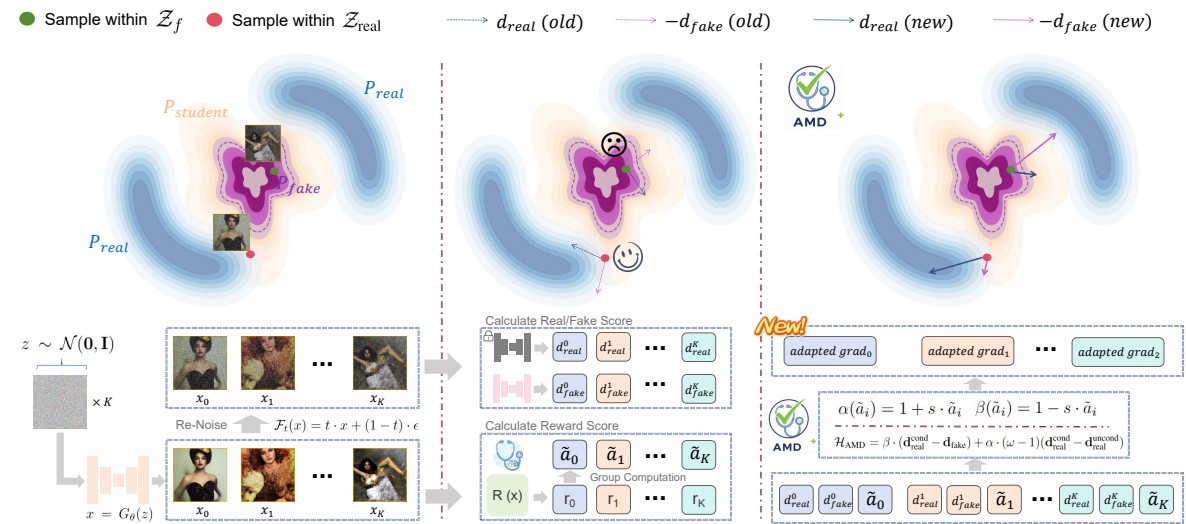

**Figure 3. Overview of AMD.** The framework operates in three stages: **(Left)** Group Generation & Re-noising: For each prompt, the student $G_\theta$ produces a group of samples $\{x_i\}_{i=1}^{K}$, which are subsequently perturbed by the forward operator $\mathcal{F}_t$. **(Middle)** Reward-aware Diagnosis: A reward model $R(\cdot)$ serves as a proxy to pinpoint samples (e.g., $x_K$) trapped in the *Forbidden Zone* ($\mathcal{Z}_f$), where the real teacher's score $s_{\text{real}}$ becomes ill-posed. **(Right)** Dynamic Score Adaption: Through the adaptive operator $\mathcal{H}_{AMD}$, AMD rectifies the combination of $\mathbf{d}_{real}$ and $\mathbf{d}_{fake}$ to derive an optimized gradient direction, thereby facilitating a rapid escape from the *Forbidden Zone* and enabling the model to surpass the teacher under reward guidance.

operator (Eq. 7) can be expanded and rearranged as follows:

$$
\begin{aligned}
\mathcal{H}_{\text{std}} &= \mathbf{d}_{\text{real}} - \mathbf{d}_{\text{fake}} \\
&= \left[\mathbf{d}_{\text{real}}^{\text{uncond}} + \omega(\mathbf{d}_{\text{real}}^{\text{cond}} - \mathbf{d}_{\text{real}}^{\text{uncond}})\right] - \mathbf{d}_{\text{fake}} \\
&= \underbrace{(\mathbf{d}_{\text{real}}^{\text{cond}} - \mathbf{d}_{\text{fake}})}_{\text{DM Term}(\mathbf{d}_{dm})} + \underbrace{(\omega - 1)(\mathbf{d}_{\text{real}}^{\text{cond}} - \mathbf{d}_{\text{real}}^{\text{uncond}})}_{\text{CA Term}(\mathbf{d}_{ca})}.
\end{aligned} \tag{9}
$$

This decomposition reveals the structural asymmetry in the gradient: the DM Term directly anchors the student to the valid data manifold, whereas the CA Term enforces semantic conditions. Consequently, to robustly escape $\mathcal{Z}_f$, a conservative update that prioritizes the DM Term is necessary; once the student resides in safer regions, the CA Term can be amplified to accelerate distillation.

Building on this insight, we define the **Dynamic Score Adaptation** strategy via

$$
\mathcal{H}_{\text{AMD}} = \beta \cdot (\mathbf{d}_{\text{real}}^{\text{cond}} - \mathbf{d}_{\text{fake}}) + \alpha \cdot (\omega - 1)(\mathbf{d}_{\text{real}}^{\text{cond}} - \mathbf{d}_{\text{real}}^{\text{uncond}}), \tag{10}
$$

where $\alpha$ and $\beta$ are dynamic coefficients adaptively modulated according to sample reliability.

**Reward as a Diagnostic Proxy** Since the energy potential $E_{\text{real}}(\cdot)$ is analytically intractable in high-dimensional space, we employ a pre-trained reward model $R(\cdot)$ as a practical proxy to identify samples residing in the *Forbidden Zone* $\mathcal{Z}_f$, which in turn informs the adaptive coefficients $\alpha$ and $\beta$.

**Assumption 3.3** (Preference–Competence Alignment). Let $\mathcal{Z}_{\text{pref}}(\tau) = \{x \mid R(x) > \tau\}$ denote the high-reward region. We assume that samples in $\mathcal{Z}_{\text{pref}}(\tau)$ are *statistically*

*concentrated* in the low-energy regime of the real teacher as

$$
\mathbb{P}\big(E_{\text{real}}(x) \le \gamma' \mid x \in \mathcal{Z}_{\text{pref}}(\tau)\big) \ge 1 - \delta, \tag{11}
$$

where $\gamma' < \gamma$ and $\delta \ll 1$.

Inspired by Matsutani et al. (2025), which shows that reward model fundamentally contracts the search space by concentrating probability mass onto a narrow, high-reward subset of the output distribution, we posit Assumption 3.3, which implies that the reward model implicitly induces a reliability ordering over the sample space. As a result, regions of low reward naturally serve as a practical proxy for $\mathcal{Z}_f$, where the real teacher's score estimates become unreliable and potentially misleading.

To eliminate prompt-dependent scale variance, we adopt a group-relative sensing strategy inspired by GRPO (Shao et al., 2024). For a group $\{x_0, \dots, x_K\}$ generated from the same prompt by student, we compute the normalized relative advantage $\tilde{a}_i$ as

$$
\tilde{a}_i = clip\left(\frac{R(x_i) - \mu_g}{\sigma_g + \epsilon}, -1, 1\right), \tag{12}
$$

where $\mu_g$ and $\sigma_g$ are the mean and standard deviation of rewards within the group. Equipped with $\tilde{a}_i$ as a diagnostic proxy for the Forbidden Zone $\mathcal{Z}_f$, we dynamically modulate the distillation weights $\alpha$ and $\beta$ in Equation (10) through a linear adaptive rule:

$$
\alpha(\tilde{a}_i) = 1 + s \cdot \tilde{a}_i, \quad \beta(\tilde{a}_i) = 1 - s \cdot \tilde{a}_i, \tag{13}
$$

where $s \in \mathbb{R}^+$ is a sensitivity hyperparameter that governs the intensity of the reactive modulation.

This formulation effectively re-scales the potential field $\mathcal{V}_{\text{DMD}}$ on a per-sample basis: for low-advantage samples ($\tilde{a}_i < 0$), the repulsive force is amplified to facilitate an active escape from $\mathcal{Z}_f$; for high-advantage samples ($\tilde{a}_i > 0$), the attractive force is prioritized to refine generation fidelity. The conceptual workflow of AMD is illustrated in Fig. 3, and the detailed algorithm is detailed in Algorithm 1.

### 3.3. Repulsive Landscape Sharpening

While the score adaption in Section 3.2 determines *how* to combine gradients, it relies on a fundamental assumption: that the fake teacher can provide a meaningful repulsive signal $\mathbf{d}_{fake}$ within the *Forbidden Zone* $\mathcal{Z}_f$.

In standard DMD (Yin et al., 2024b), the fake teacher is primarily designed to promote diversity. By uniformly modeling the student's distribution, it exerts a dispersive force that prevents mode collapse. However, our analysis in Section 3.1 assigns a critical new role to the fake teacher: **correction**. To effectively repel the student from the *Forbidden Zone*, the fake teacher must first learn to recognize it.

To address this, we introduce **Repulsive Landscape Sharpening**, which reallocates the training focus of fake teacher $\psi$ towards failure cases. We modify the distillation objective by incorporating an advantage-aware weight $\mathbf{W}(\tilde{a}_i)$ as

$$\mathcal{L}_\psi = \mathbb{E}_{z,t,\epsilon}\left[\mathbf{W}(\tilde{a}_i) \cdot \|\epsilon - \psi(x_t, t)\|^2\right], \qquad (14)$$

where $x_t = \mathcal{F}_t(G_\theta(z))$ and $\mathbf{W}(\cdot)$ imposes a heavier penalty on low-advantage samples (e.g., $\mathbf{W}(\tilde{a}_i) = e^{-\tilde{a}_i}$).

This mechanism ensures that the fake teacher learns a highly sensitive energy landscape specifically tailored to the student's current weaknesses. Mathematically, this creates a steep likelihood slope, maximizing the magnitude of the repulsive gradient $\mathbf{d}_{\text{fake}} \propto \nabla \log p_{\text{fake}}$ during the student's update. Thus, the fake teacher transforms from a passive density estimator into an active failure detector, providing the "necessary kick" to escape the Forbidden Zone.

## 4. Experiments

In this section, we conduct extensive experiments to validate AMD. As a natural corollary of the framework in Section 3.1, these experiments primarily serve to verify the rationality of our theoretical analysis.

### 4.1. Experiments Setting

We evaluate AMD across a diverse range of diffusion backbones, modalities, and architectural designs. For image generation, we conduct experiments on SiT, and SDXL (Podell et al., 2023); for video generation, we evaluate AMD on Wan2.1 (Wan et al., 2025) and LongLive (Yang et al., 2025a). To ensure comprehensive coverage, we report results on mul-

tiple benchmarks, including MS-COCO (Lin et al., 2015), ImageNet (Russakovsky et al., 2015), DrawBench (Saharia et al., 2022), HPD (Wu et al., 2023), GenEval (Ghosh et al., 2023), VBench (Huang et al., 2023), VBench++ (Huang et al., 2025c), VideoGen-Eval (Yang et al., 2025b) and TA-Hard (Liu et al., 2025c). Regarding the reward models, we utilize HPSv2 (Wu et al., 2023) for SDXL, DINOv2 (Caron et al., 2021) for DiT-based models (SiT) (Ma et al., 2024), and VideoAlign (Liu et al., 2025c) for video models. Detailed configurations are provided in Section A.

### 4.2. Main Results

**Image Generation** We first conduct experiments on text-to-image generation. We follow standard evaluation protocols and first assess performance on the 10k COCO2014-val dataset. As presented in Tab. 3, our method significantly outperforms existing state-of-the-art acceleration techniques. Specifically, AMD achieves the highest ImageReward (88.37) and HPSv2 (31.25), surpassing strong competitors like PCM and D-DMD. To further verify the robustness of our method, we extend our evaluation to multiple popular benchmarks. As shown in Tab. 7, AMD consistently surpasses DMD2 across all stylistic categories (Anime, Photo, Concept-art, and Painting) on the HPDv2 dataset. Additionally, on the object-focused GenEval benchmark (Tab. 2), AMD secures the top rank among distilled models with an overall score of 0.57, exhibiting superior prompt adherence (please refer to Tab. 8 for detailed scores of each dimension). **It is worth noting that since the official weights for DMDR and D-DMD are not publicly available, we reference the results directly from their respective papers for comparison.**

*Table 2.* Comparison of different text-to-image models on GenEval (Ghosh et al., 2023) benchmark. (base model: SDXL) The best results in each category are highlighted in **bold**.

| Method | #Params | Resolution | NFEs | Overall ↑ |
|---|---|---|---|---|
| *Pretrained Models* | | | | |
| SDXL (Podell et al., 2023) | 2.6B | $1024 \times 1024$ | $50 \times 2$ | 0.55 |
| FLUX.1-dev (Labs, 2024) | 12.0B | $1024 \times 1024$ | 50 | 0.66 |
| *Pretrained Models* | | | | |
| SDXL-LCM (Luo et al., 2023) | 2.6B | $1024 \times 1024$ | 4 | 0.50 |
| SDXL-Turbo (Podell et al., 2023) | 2.6B | $512 \times 512$ | 4 | 0.56 |
| SDXL-Lightning (Lin et al., 2024) | 2.6B | $1024 \times 1024$ | 4 | 0.53 |
| SDXL-DMD2 (Yin et al., 2024a) | 2.6B | $1024 \times 1024$ | 4 | 0.51 |
| SDXL-DMDR (Jiang et al., 2025) | 2.6B | $1024 \times 1024$ | 4 | 0.56 |
| SDXL-AMD (Ours) | 2.6B | $1024 \times 1024$ | 4 | **0.57** |

Furthermore, we provide a human preference winning rate analysis in Fig. 9. The results indicate that AMD holds a substantial lead over DMD2 on both DrawBench and HPDv2, confirming that our method generates images with superior quality. Due to the page limit, we provide additional results and visualization in Section F.2 and Section F.3.

*Table 3.* Quantitative comparison of text-to-image task on 10k COCO2014-val prompts (Lin et al., 2015). (base model: SDXL). The results validate the effectiveness of our proposed AMD.

| Method | ImageReward ↑ | HPSv2 ↑ |
|---|---|---|
| LCM (Luo et al., 2023) | 39.56 | 28.00 |
| Turbo (Sauer et al., 2024) | 46.09 | 29.83 |
| Lightning (Lin et al., 2024) | 57.48 | 30.30 |
| Flash (Chadebec et al., 2025) | 19.04 | 27.71 |
| PCM (Wang et al., 2024) | 64.73 | 30.76 |
| DMD2 (Yin et al., 2024a) | 71.01 | 30.64 |
| D-DMD (Liu et al., 2025a) | 78.61 | 30.34 |
| AMD (Ours) | **88.37** | **31.25** |

*Table 4.* Quantitative comparison on 50K-ImageNet (256 × 256). (base model: SiT-XL/2). Notably, DMDR exhibits reward hacking behavior, attaining high IS at the cost of poor FID. AMD, however, shows a superior balance between image quality and diversity.

| Method | FID ↓ | sFID ↓ | IS ↑ |
|---|---|---|---|
| DMD | 3.5573 | 5.8499 | 314.42 |
| DMDR | 9.6341 | 7.3541 | **391.79** |
| AMD (Ours) | **3.4690** | **5.7464** | 316.02 |

To further isolate the intrinsic effectiveness of our distillation mechanism, we conduct experiments on the class-to-image generation on ImageNet. Using SiT-XL/2 (Ma et al., 2024) as the backbone, we compare our proposed **AMD** against standard Standard DMD and DMDR (Jiang et al., 2025). The implementation details are provided in Section A.4. As shown in Tab. 4, AMD achieves superior FID (3.4690) and sFID (5.7464) to DMD, demonstrating its capacity to sculpt the generation manifold rather than merely matching the teacher. Unlike DMDR, which suffers from severe mode collapse, AMD's tempered refinement provides a balanced optimization path that avoids reward hacking while steadily enhancing overall quality.

**Video Generation** We extend AMD to streaming video generation task using Wan2.1-1.3B as the backbone, following the first stage of LongLive (Yang et al., 2025a) protocol. As shown in Tab. 5, AMD surpasses the LongLive baseline with a Total Score of **82.21 vs. 81.42**. Tab. 6 further reveals substantial gains on VBench, where AMD boosts Motion Quality by ∼67% (35.51 → 59.26) and the Total Score to 197.45. The qualitative comparisons are shown in Fig. 4.

While Visual and Motion Quality improve significantly, the slight trade-off in Text Alignment (TA) is an anticipated outcome of the VideoAlign reward, which explicitly prioritizes dynamic fidelity and motion aesthetics over rigid semantic constraints. To further demonstrate the scalability of our approach, we provide additional results using the larger Wan2.1-14B backbone in Section F.2, confirming that

*Table 5.* Quantitative comparison on the streaming video generation task. Base model: Wan-1.3B. The results on VBench show that AMD outperforms the baseline in terms of Total Score.

| Model | Evaluation scores ↑ | | |
|---|---|---|---|
| | Total | Quality | Semantic |
| LTX-Video (HaCohen et al., 2024) | 80.00 | 82.30 | **70.79** |
| MAGI-1 (Teng et al., 2025) | 79.18 | 82.04 | 67.74 |
| CausVid (Yin et al., 2025) | 81.20 | 84.05 | 69.80 |
| LongLive (Yang et al., 2025a) | 81.42 | 84.20 | 70.38 |
| AMD (Ours) | **82.21** | **85.29** | 69.91 |

*Table 6.* Quantitative comparison on the streaming video generation task. (base model: Wan-1.3B). We compare AMD against the LongLive using preference-based metrics. The results demonstrate that AMD achieves significant improvements in Visual Quality (VQ) and Motion Quality (MQ), leading to higher overall scores.

| Dataset | Model | VQ ↑ | MQ ↑ | TA ↑ | Total ↑ |
|---|---|---|---|---|---|
| VBench (Huang et al., 2023) | LongLive | 30.10 | 35.51 | **107.98** | 173.59 |
| | AMD (Ours) | **37.39** | **59.26** | 100.79 | **197.45** |
| VideoGen-Eval (Yang et al., 2025b) | LongLive | -24.53 | -13.07 | **118.57** | 80.96 |
| | AMD (Ours) | **-21.93** | **17.87** | 107.98 | **87.84** |
| TA-Hard (Liu et al., 2025c) | LongLive | -0.9908 | 13.19 | **27.18** | 39.39 |
| | AMD (Ours) | **-0.1412** | **28.74** | 14.65 | **43.52** |

AMD's effectiveness persists across model scales.

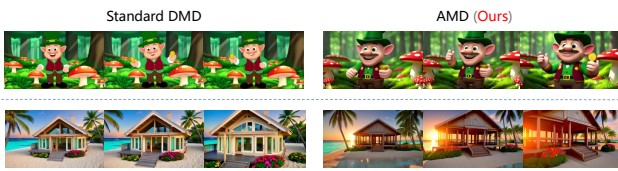

*Figure 4.* Qualitative comparison on text-to-video generation. Compared to standard DMD (e.g., LongLive), AMD demonstrates superior visual fidelity and more coherent motion realism. Additional qualitative results are provided in Fig. 15.

### 4.3. Ablation Study

**Effectiveness of AMD Components.** AMD achieves comprehensive gains through two synergistic mechanisms: (i) **Dynamic Score Adaptation** provides fine-grained modulation of DMD updates, adaptively refining both the magnitude and direction of the distillation gradients based on the sample state. (ii) **Repulsive Landscape Sharpening** heightens the system's perception of failure regions, sharpening repulsive forces to more effectively identify and rectify pathological areas.

As shown in Figure 5, we observe that FID and IS improve alongside rising reward scores throughout training, further validating the robustness and effectiveness of our optimization objective.

**Training Dynamics and Stability.** Here, we analyze the training dynamics of AMD. As shown in Fig. 6, we track both Inception Score (IS) and DINO-based reward during

*Table 7.* Quantitative comparison of text-to-image task on HPDv2 (Wu et al., 2023) dataset. We compare AMD against DMD2 using a lot of preference-based metrics. Base model: SDXL.

| Method | Anime | | | Photo | | | Concept-art | | | Painting | | | Average | | |
|---|---|---|---|---|---|---|---|---|---|---|---|---|---|---|---|
| | PickScore ↑ | HPSv2 ↑ | ImageReward ↑ | PickScore ↑ | HPSv2 ↑ | ImageReward ↑ | PickScore ↑ | HPSv2 ↑ | ImageReward ↑ | PickScore ↑ | HPSv2 ↑ | ImageReward ↑ | PickScore ↑ | HPSv2 ↑ | ImageReward ↑ |
| DMD2 | 22.85 | 33.16 | 120.74 | 22.41 | 30.31 | 71.20 | 22.03 | 31.39 | 106.76 | 22.18 | 31.72 | 117.67 | 22.36 | 31.64 | 104.09 |
| AMD (Ours) | **22.91** | 33.10 | 116.20 | **22.96** | **30.76** | **84.01** | **22.60** | **32.04** | **113.49** | **22.40** | **31.96** | 108.37 | **22.72** | **31.97** | **105.52** |

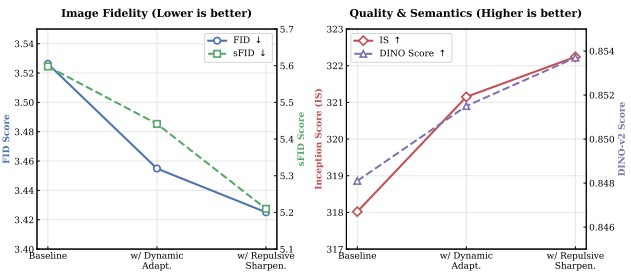

*Figure 5.* Component-wise ablation study on 50K-ImageNet (256×256). (base model: SiT-XL/2). We investigate the contribution of each component in AMD. Dynamic Adapt denotes the Dynamic Score Adaptation (Section 3.2), and Repulsive Sharpen denotes the Repulsive Landscape Sharpening (Section 3.3).

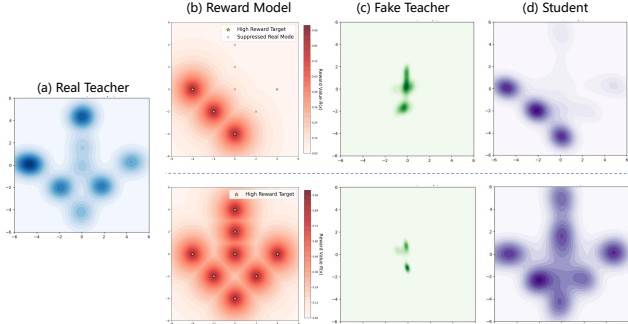

*Figure 7.* Reward-Driven Shaping. **Top:** Under a selective reward (b), the Student (d) ignores the fixed Teacher's (a) gradients in suppressed zones (×) to strictly match high-value modes (⋆). **Bottom:** Global alignment recovers the full distribution.

distillation on SiT-XL/2. Compared with the baseline, AMD demonstrates consistently faster improvement and a more stable training trajectory across iterations. Notably, the increase in reward is well aligned with the improvement in IS, indicating that reward-aware dynamic score adaptation provides reliable diagnostic signals and effectively steers the student away from low-quality regions during training, rather than merely improving the final outcome.

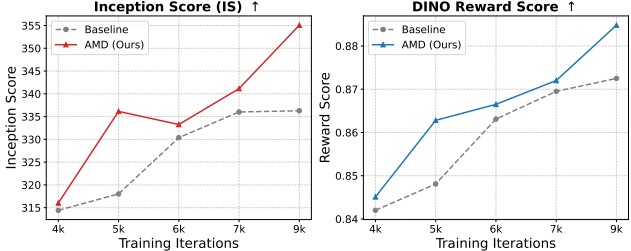

*Figure 6.* Reward–Quality Co-evolution during Distillation. AMD exhibits a synchronized increase in generation quality (IS) and reward scores, suggesting that reward-aware dynamic score adaptation effectively guides the student out of low-quality regions.

**Impact of Reward Model.** To intuitively validate the reward model's role as a diagnostic proxy, we visualize training dynamics on a 2D multi-modal dataset in Fig. 7. We maintain a fixed Real Teacher (Col. a) while varying the reward landscape (Col. b).

In the Selective Guidance scenario (**Top Row**), where specific modes are suppressed by the reward model (marked ×), the Student (Col. d) successfully converges only to high-reward targets, effectively overriding the Real Teacher's attractive gradients in these low-value regions.

In contrast, under Global Alignment (**Bottom Row**), the stu-

dent faithfully recovers the full distribution. This confirms that AMD effectively leverages the reward signal to dynamically reshape the generation manifold, ensuring the student avoids *Forbidden Zones* even when they are supported by the original teacher. By actively sculpting the student's distribution according to the reward landscape, AMD enables the student to selectively learn useful modes and effectively **transcend the performance ceiling** of the original teacher.

## 5. Conclusion

In this work, we identify *Forbidden Zones* as a fundamental failure regime in Distribution Matching Distillation (DMD), where unreliable guidance from the real teacher and vanishing repulsion from the fake teacher jointly stall optimization. To address this, we propose **Adaptive Matching Distillation (AMD)**. Empirically, AMD demonstrates robust effectiveness across both image and video generation benchmarks, successfully bridging the gap between perceptual quality and distributional diversity while overcoming the inherent limitations of the original teacher.

More broadly, our framework provides a unified optimization perspective on DMD-style methods, revealing them as implicit strategies for navigating corrupted regions. Due to space constraints, we discuss the limitations of our approach and outline future research directions in Section D. By reframing distillation as a latent navigation task, AMD offers the community a novel lens to diagnose and rectify the inherent trade-offs in distribution matching. We hope this perspective encourages the development of more adaptive and self-correcting distillation paradigms that move beyond static alignment toward active generative refinement.

# Impact Statement

We present ADM (Adaptive Matching Distillation), a framework designed for efficient and high-fidelity few-step generation via reward-aware distribution matching distillation. The training process of AMD relies on guidance from pretrained teacher models and reward models. While our experiments utilize standard academic datasets, we acknowledge that the safety and ethical alignment of the distilled student model are intrinsically tied to the quality of these foundational signals. Therefore, we emphasis on the responsible deployment of this technology. We advocate for the integration of content authentication mechanisms and strive to maximize the societal benefits of efficient generation while diligently mitigating potential risks associated with misuse, such as the creation of misleading deepfake content.

# Acknowledgements

This work was supported by the National Natural Science Foundation of China under Grant No. 62506317 and Guangdong Provincial Key Lab of Integrated Communication, Sensing and Computation for Ubiquitous Internet of Things (No.2023B1212010007). This work was supported by National Major S&T Special Project on New Generation Artificial Intelligence (Nos. 2025ZD0123500), National Natural Science Fund of China (Nos. 62506155), Provincial Natural Science Fund of Jiangsu (Nos. BK20251985), and Suzhou Municipal Leading Talents Fund (Nos. ZXL2025320).

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

# A. Implementation Details

In this section, we provide an overview of the benchmarks, evaluation metrics, diffusion models and the hyperparameter settings in our main paper.

## A.1. Benchmark

**MS-COCO.** To assess zero-shot text-to-image synthesis performance, we utilize the MS-COCO 2014 validation set (Lin et al., 2015). Following the evaluation protocols established in prior works like DMD (Yin et al., 2024b) and DMD2 (Yin et al., 2024a), we adopt two distinct evaluation settings. First, to measure distributional fidelity, we generate 30,000 images (COCO-30k) using randomly sampled prompts from the validation set. These samples are resized to $256 \times 256$ resolution and compared against the full reference set of 40,504 real images to compute FID scores (Heusel et al., 2018) using the clean-fid library (Parmar et al., 2022). Second, for assessing text-image alignment and fine-grained details, we utilize a subset of 10,000 prompts (COCO-10k). In this setting, we generate corresponding images at $512 \times 512$ resolution and compare them directly with their specific ground-truth references to calculate ImageReward and HPSv2 scores.

**ImageNet.** We evaluate class-conditional image generation capabilities using the ImageNet benchmark (Russakovsky et al., 2015) at resolutions of $256 \times 256$ and. Following the rigorous protocols established in SiT (Ma et al., 2024), we assess generation fidelity using the Fréchet Inception Distance (FID-50K). Specifically, we generate 50,000 samples and compute metrics against the statistics of the entire ImageNet training set, rather than the validation split. To ensure strict consistency and fair comparison with baseline diffusion models.

**DrawBench.** To evaluate the model's robustness across diverse semantic challenges, we utilize DrawBench (Rombach et al., 2022)[1]. This challenging benchmark comprises 200 carefully curated text prompts, organized into 11 distinct categories. These categories are specifically designed to probe difficult generation capabilities, such as counting, spatial reasoning, rare object generation, and the handling of long, complex descriptions. By covering such a wide spectrum of linguistic and visual difficulties, DrawBench serves as a fine-grained diagnostic tool for identifying specific limitations in T2I synthesis.

**HPD v2.** We also employ the Human Preference Dataset v2 (HPD v2) (Wu et al., 2023) is an extensive dataset featuring clean and precise annotations. With 798,090 binary preference labels across 433,760 image pairs, it addresses the limitations of conventional evaluation metrics that fail to accurately reflect human preferences. Following the methodologies in (Wu et al., 2023), we employed four distinct subsets for our analysis: Animation, Concept-art, Painting, and Photo, each containing 800 prompts.

**GenEval.** For a quantitative assessment of compositional fidelity, we adopt the GenEval framework (Ghosh et al., 2023). This benchmark evaluates how well generated images adhere to prompt constraints regarding object co-occurrence, spatial arrangement, counting, and color binding. GenEval automates this process by utilizing pre-trained object detection models to verify visual attributes, serving as a scalable proxy for human judgment. The test set consists of 550 concise and unambiguous prompts, allowing for a focused analysis of the model's ability to execute precise visual instructions.

**VBench and VBench++.** VBench (Huang et al., 2023) serves as our primary framework for assessing video generation performance. This comprehensive suite moves beyond single-value metrics by decomposing video quality into 16 hierarchical and disentangled dimensions, such as subject consistency, motion smoothness, temporal flickering, and aesthetic quality. For each dimension, VBench utilizes a tailored prompt suite and specific evaluation protocols to isolate and measure individual model capabilities. In addition, Huang et al. (2025c) introduce VBench++, which extends the benchmark to support image-to-video evaluation. VBench++ incorporates an adaptive Image Suite that enables fair and consistent assessment across diverse generation settings.

**VideoGen-Eval.** To address the spatiotemporal complexity of modern video synthesis, we employ VideoGen-Eval (Yang et al., 2025b). This agent-based framework moves beyond static metrics by utilizing a dynamic evaluation pipeline. It begins with 700 structured, content-rich prompts that explicitly define components such as camera movement, lighting, and subject interactions. An LLM acts as a content structurer to decompose these prompts, while a Multimodal Large Language

---

[1]https://huggingface.co/datasets/shunk031/DrawBench

Model (MLLM), augmented with specialized temporal patch tools—serves as the adjudicator. This hierarchical approach allows for a granular assessment of how well generated videos adhere to multifaceted instructions, ensuring high alignment with human judgment across a diverse set of generated assets.

**TA-Hard.** Standard benchmarks often utilize prompts that are semantically straightforward, potentially masking a model's limitations in complex instruction following. To mitigate this, we incorporate the TA-Hard prompt set (Liu et al., 2025c). This collection is specifically curated to stress-test Text Alignment (TA) by focusing on high semantic density and imaginative scenarios, such as anthropomorphic animals playing instruments or complex interactions between disparate objects. By presenting the model with intricate and counter-intuitive descriptions that require precise compositional reasoning, TA-Hard serves as a rigorous diagnostic tool for evaluating the upper bounds of a model's ability to maintain fidelity to challenging textual constraints.

## A.2. Evaluation Metric

**PickScore.** PickScore (Kirstain et al., 2023) serves as a preference-based evaluation metric trained on the massive Pick-a-Pic dataset. By fine-tuning a CLIP model to predict human choices from pairwise comparisons, it effectively captures the nuances of user satisfaction. Unlike distributional metrics such as FID, PickScore is designed to evaluate individual sample quality and alignment, offering a stronger correlation with human judgment in open-domain text-to-image generation tasks compared to traditional benchmarks.

**HPS v2.** The human preference score version 2 (HPS v2) is an improved model to predict user preferences, created by fine-tuning the CLIP model (Radford et al., 2021) on the HPD v2. This refined metric is designed to align T2I generation outputs with human tastes by estimating the likelihood that a synthesized image will be preferred, thereby serving as a reliable benchmark for evaluating the performance of T2I models across diverse image distributions.

**ImageReward.** ImageReward (Xu et al., 2023) is a comprehensive reward model trained via reinforcement learning from human feedback (RLHF) principles. By learning from a vast collection of expert-annotated comparisons, it addresses the limitations of standard metrics by mitigating "gamification" issues. It jointly assesses fidelity and alignment, demonstrating a correlation with human ranking that surpasses traditional metrics like IS and FID, establishing it as a robust tool for automated evaluation.

**VideoAlign.** VideoAilgn (Liu et al., 2025c) is a multi-dimensional reward model designed for assessing video generation. It is trained on the *VideoGen-RewardBench* dataset, a large-scale collection of 26.5k video triplets with human preference annotations. Unlike typical reward models that focus on a single quality metric, VideoAlign evaluates videos across three distinct dimensions: **Visual Quality (VQ)**, **Motion Quality (MQ)**, and **Text Alignment (TA)**. These scores can be aggregated into a total reward that reflects overall human preference. The model is built on a Vision-Language Model (VLM) backbone (e.g., Qwen2-VL-2B) and trained using a Bradley-Terry model with ties (BTT) objective, making it robust to tied preferences and effective for aligning flow-based video generation models.

## A.3. Diffusion Models

In the main paper, we totally use 5 diffusion models, including SiT for class-to-image generation, SD1.5 and SDXL for text-to-image generation, and LongLive and Wan2.1 for text-to-video generation.

**SiT.** Scalable Interpolant Transformers (SiT), proposed by Ma et al. (2024), are a family of generative models built upon the Diffusion Transformer (DiT) backbone. SiT utilizes the stochastic interpolant framework, which allows for a more flexible connection between the data and noise distributions compared to standard diffusion models. This framework enables a modular exploration of critical design choices, including discrete versus continuous time learning, different model prediction targets (e.g., velocity vs. score), and various interpolant paths. By adopting a velocity-prediction parameterization and linear interpolants, SiT achieves superior performance over standard DiTs on ImageNet benchmarks with identical model size and compute.

**SDXL.** SDXL (Podell et al., 2023) is a latent diffusion model that significantly scales up the U-Net backbone to approximately 2.6B parameters. To improve text comprehension and prompt adherence, it employs a dual text encoder

architecture, combining the outputs of CLIP ViT-L and OpenCLIP ViT-bigG. SDXL addresses common training data issues through novel micro-conditioning schemes, where the model is conditioned on the original image resolution and cropping coordinates, allowing it to generate high-quality images without discarding small or non-square training samples. Additionally, it utilizes multi-aspect training and a separate refinement model to enhance the visual fidelity of high-resolution synthesis.

**LongLive.**    LongLive (Yang et al., 2025a) is a frame-level autoregressive framework designed for real-time, interactive long video generation. Built by fine-tuning the Wan2.1-1.3B model, LongLive adopts a causal attention mechanism that leverages KV caching to accelerate inference. To enable smooth transitions during interactive prompt switching, it introduces a *KV-recache* technique that updates cached states with new prompt embeddings. Furthermore, LongLive employs a *streaming long tuning* strategy to align training with long-context inference, alongside a *short window attention* mechanism paired with a *frame-level attention sink*. These designs allow the model to maintain long-range temporal consistency and high throughput, achieving approximately 20 FPS on a single H100 GPU for videos up to 240 seconds.

**Wan2.1.**    Wan2.1, introduced in (Wan et al., 2025), is an open-source video generation model developed by Alibaba, based on a Diffusion Transformer (DiT) architecture and flow matching framework. It supports multiple tasks including text-to-video (T2V) and image-to-video (I2V). The model is available in two versions: a 14B-parameter variant for high-quality 720p generation and a lightweight 1.3B variant suitable for consumer-grade GPUs. Due to the resource limits, in this paper, we utilize the Wan2.1-T2V-14B.

### A.4. Hyperparameter Settings

Here we provide the detailed hyperparameter configurations for both image and video generation tasks.

**Video Generation.**    For the **streaming video generation** task, we strictly follow the first-stage experimental protocol of LongLive (Yang et al., 2025a). We utilize Wan2.1-1.3B (Wan et al., 2025) as the base model and conduct distillation for 700 iterations with a learning rate of $2.0 \times 10^{-6}$ and a total batch size of 8. Specifically, the local attention size is set to 12, and the frame sink size is 3. For **bidirectional-attention video generation**, we employ Wan2.1-14B as the backbone, performing 800 distillation steps with a learning rate of $5.0 \times 10^{-7}$ and a batch size of 8. Across all video tasks, we adopt VideoAlign (Liu et al., 2025c) as the reward proxy. During evaluation, we generate videos with 81 frames at 16 FPS, corresponding to a duration of 5 seconds.

**Image Generation.**    For class-to-image generation, we utilize SiT-XL/2 (Ma et al., 2024) as the backbone for all ImageNet (256×256) experiments. For the main results in Tab. 4, we adopt a **two-stage training protocol**: all models are first initialized from a Pure-DMD checkpoint pre-trained for 3,000 iterations. Subsequently, we perform fine-tuning using Pure-DMD, DMDR, and AMD, respectively, for an additional 1,000 iterations. For the ablation studies in Tab. **??**, the second-stage fine-tuning is extended to 2,000 iterations to ensure full convergence for component analysis. The total batch size is 512, and the learning rate is set to $2.0 \times 10^{-5}$ with DINOv2 (Caron et al., 2021) serving as the reward proxy. For experiments on text-to-image, especially for SDXL, as shown in Tab. 7, Tab. 3, Tab. 9, and Tab. 2, we strictly follow the training setting in DMD2 (Yin et al., 2024a). However, due to the resource limit, we only adopt 8 H800 to distill the model.

## B. Related Work

In this section, we review existing works relevant to AMD.

**Distribution Distillation in Diffusion Model**    Diffusion models are commonly distilled into low-step student models to reduce inference cost. A line of works studies *score distillation*, where a pretrained diffusion model provides score-based supervision for training fast generators. Score distillation was first popularized in text-to-3D generation (Hertz et al., 2023; Poole et al., 2022), where a pretrained text-to-image diffusion model provides score-based supervision for optimizing 3D representations. Building upon this idea, Yin et al. (2024b) proposed *Distribution Matching Distillation (DMD)*, which minimizes an approximate KL divergence between the student and teacher distributions. Subsequent work, DMD2 Yin et al. (2024a), further introduces an adversarial (GAN) loss to improve training stability and sample diversity.

More recent variants (Jiang et al., 2025; Shao et al., 2025; Liu et al., 2025a) focus on improving the robustness and stability of the DMD objective. Decoupled-DMD (Liu et al., 2025a) reformulates the objective into a CFG-augmented matching

term and a regularization term, together with a more stable renoising scheduler. MagicDistillation (Shao et al., 2025) argues that, beyond updating the fake teacher, the real teacher should also be adaptively updated during training, while DMDR (Jiang et al., 2025) introduces an auxiliary reinforcement learning objective to optimize the student independently. In addition, several works (Huang et al., 2025b; Yang et al., 2025a; Liu et al., 2025d) extend the DMD paradigm to long video generation.

In this work, we provide a unified perspective on DMD variants and show that their modifications can be viewed as implicit strategies for avoiding unreliable regions of the real teacher. We formalize such regions as *Forbidden Zones*, where naive score matching leads to unstable or misleading gradients.

**Optimization View on Generation Tasks.**   Generative modeling can be fundamentally viewed as an optimization problem aimed at finding high-probability modes within a complex energy landscape. Sampling methods, such as Langevin Dynamics (Song & Ermon, 2019) and Probability Flow ODEs (Song et al., 2020), essentially perform gradient ascent on the log-density score field to traverse from noise to data. This optimization perspective has been explicitly leveraged in controllable generation, where external guidance signals—such as classifier gradients (Dhariwal & Nichol, 2021; Bai et al., 2025) or CLIP scores (Radford et al., 2021)—are injected into the sampling process to steer trajectories toward desired attributes (Bansal et al., 2023; Liu et al., 2023a). Our work aligns with this philosophy but operates at the training stage: **we treat the distillation process as navigating a treacherous energy field, requiring dynamic gradient reshaping to traverse *Forbidden Zones*.**

**Reward-Guided Diffusion Training.**   The integration of Reinforcement Learning (RL) into diffusion training has gained significant traction. Recent methods (Liu et al., 2025b; Wu et al., 2025b; Zheng et al., 2025; He et al., 2025; Liu et al., 2026) treat the denoising chain as a policy, directly optimizing it to maximize global rewards. Most pertinent to our work is the recently proposed **Reward Forcing** (Lu et al., 2025), which biases the student's distribution by prioritizing training samples with high reward scores. However, a critical distinction separates AMD from such prioritization schemes. While Reward Forcing focuses on emphasizing *where to go* by up-weighting successful samples, AMD focuses on *how to escape* when the model fails. We treat low-reward samples not merely as data to be down-weighted, but as active indicators of *Forbidden Zones*.

## C. Proof

### C.1. Proof of Proposition 3.1

*Proof.* Our goal is to show that the parameter update of the student $G_\theta$ induces an effective gradient descent trajectory in the latent space. We proceed in three steps: relating parameter shifts to latent shifts, converting scores to denoised targets, and identifying the resulting potential function.

**Step 1: First-order Approximation of Latent Update.**   Consider the infinitesimal update of the student parameters: $\Delta\theta = -\eta\nabla_\theta\mathcal{L}_{\text{DMD}}$. For a fixed latent code $z$, the change in the generated sample $x = G_\theta(z)$ can be approximated via the first-order Taylor expansion:

$$x_{new} - x \approx \frac{\partial G_\theta(z)}{\partial\theta}\Delta\theta = -\eta\frac{\partial G_\theta(z)}{\partial\theta}\nabla_\theta\mathcal{L}_{\text{DMD}}. \tag{15}$$

Substituting the DMD gradient formulation from Eq. 2, we obtain:

$$x_{new} - x \approx \eta \cdot \mathbb{E}_t\left[\left(s_{\text{real}}(\mathcal{F}_t) - s_{\text{fake}}(\mathcal{F}_t)\right)\mathbf{J}_\theta\mathbf{J}_\theta^\top\right], \tag{16}$$

where $\mathbf{J}_\theta = \frac{\partial G_\theta(z)}{\partial\theta}$ is the Jacobian. In the local neighborhood of optimization, the term $\mathbf{J}_\theta\mathbf{J}_\theta^\top$ acts as a positive semi-definite scaling matrix. For the purpose of trajectory analysis, we absorb this architectural dependency into the effective step size $\eta_{\text{eff}}$, yielding:

$$x_{new} - x \propto s_{\text{real}}(\mathcal{F}_t) - s_{\text{fake}}(\mathcal{F}_t). \tag{17}$$

**Step 2: Score-to-Displacement Transformation.**   According to Tweedie's formula, the score function $s(\mathcal{F}_t)$ at diffused state $\mathcal{F}_t(x)$ is related to the denoised estimate $\hat{x}_0$ by $s(\mathcal{F}_t) = \frac{\alpha_t\hat{x}_0 - \mathcal{F}_t}{\sigma_t^2}$. The difference between the real and fake teacher

scores can thus be expressed as:

$$
\begin{aligned}
s_{\text{real}} - s_{\text{fake}} &= \frac{\alpha_t \hat{x}_{0,\text{real}} - \mathcal{F}_t}{\sigma_t^2} - \frac{\alpha_t \hat{x}_{0,\text{fake}} - \mathcal{F}_t}{\sigma_t^2} \\
&= \frac{\alpha_t}{\sigma_t^2} \left( \hat{x}_{0,\text{real}} - \hat{x}_{0,\text{fake}} \right).
\end{aligned}
\tag{18}
$$

By defining the latent displacements $\mathbf{d}_{\text{real}} = x - \hat{x}_{0,\text{real}}$ and $\mathbf{d}_{\text{fake}} = x - \hat{x}_{0,\text{fake}}$, we can rewrite the target difference as:

$$
\hat{x}_{0,\text{real}} - \hat{x}_{0,\text{fake}} = (x - \mathbf{d}_{\text{real}}) - (x - \mathbf{d}_{\text{fake}}) = -(\mathbf{d}_{\text{real}} - \mathbf{d}_{\text{fake}}).
\tag{19}
$$

**Step 3: Identification of the Contrastive Potential.** Substituting the results from Step 2 back into Eq. 17, the latent update rule becomes:

$$
x_{new} = x - \eta_{\text{eff}} \left( \mathbf{d}_{\text{real}} - \mathbf{d}_{\text{fake}} \right).
\tag{20}
$$

We observe that the term $(\mathbf{d}_{\text{real}} - \mathbf{d}_{\text{fake}})$ is exactly the gradient of the potential function $\mathcal{V}_{\text{DMD}}(x) = \frac{1}{2}\|x - \hat{x}_{0,\text{real}}\|^2 - \frac{1}{2}\|x - \hat{x}_{0,\text{fake}}\|^2$ with respect to $x$:

$$
\nabla_x \mathcal{V}_{\text{DMD}}(x) = (x - \hat{x}_{0,\text{real}}) - (x - \hat{x}_{0,\text{fake}}) = \mathbf{d}_{\text{real}} - \mathbf{d}_{\text{fake}}.
\tag{21}
$$

Thus, the latent update follows $x_{new} = x - \eta_{\text{eff}} \nabla_x \mathcal{V}_{\text{DMD}}(x)$, which characterizes a gradient descent step on the contrastive potential $\mathcal{V}_{\text{DMD}}(x)$. $\qquad\square$

## C.2. Theoretical Analysis of Prior Art

In this subsection, we provide theoretical justifications for re-interpreting **D-DMD** and **MagicDistillation** within our optimization framework. We demonstrate that both methods essentially serve to mitigate the gradient vanishing or hallucination problem within the *Forbidden Zone* $\mathcal{Z}_f$, albeit through distinct mechanisms: D-DMD via *noise-induced support expansion*, and MagicDistillation via *manifold bridging*.

**D-DMD: Risk Reduction via Noise-Induced Support Expansion.** D-DMD (Liu et al., 2025a) modifies the distillation objective by computing the real teacher's score at a higher noise level than the student's current state. Let $t_G$ be the student's generation step and $t_\phi$ be the teacher's score estimation step, where $t_\phi > t_G$.

**Proposition C.1.** *Increasing the diffusion time $t$ expands the effective support of the teacher's distribution, thereby reducing the probability that a student sample falls into the Forbidden Zone $\mathcal{Z}_f$.*

*Proof.* Let $p_0(x)$ be the clean data distribution. The distribution at time $t$, $p_t(x)$, is the convolution of $p_0$ with a Gaussian kernel $\mathcal{N}(0, \sigma_t^2 \mathbf{I})$:

$$
p_t(x) = \int p_0(y) \mathcal{N}(x; y, \sigma_t^2 \mathbf{I}) dy.
\tag{22}
$$

The *Forbidden Zone* at time $t$, denoted $\mathcal{Z}_f^{(t)}$, is formally defined as the region where the probability density is insufficient for reliable score estimation: $\mathcal{Z}_f^{(t)} = \{x \mid p_t(x) < \epsilon\}$.

Since convolution with a Gaussian is a smoothing operation, the support of $p_t(x)$ effectively widens as $\sigma_t$ increases. For a distorted student sample $x_{\text{bad}}$ that lies far from the data manifold (i.e., $x_{\text{bad}} \in \mathcal{Z}_f^{(t_G)}$), the density $p_{t_G}(x_{\text{bad}})$ vanishes, leading to an undefined or oscillating score $s_{\text{real}}(x_{\text{bad}}, t_G)$.

However, by increasing the noise scale to $t_\phi > t_G$, the Gaussian kernel broadens, ensuring that $p_{t_\phi}(x_{\text{bad}}) \gg \epsilon$. This effectively removes $x_{\text{bad}}$ from the Forbidden Zone of the higher noise level, i.e., $x_{\text{bad}} \notin \mathcal{Z}_f^{(t_\phi)}$. Consequently, the D-DMD update provides a valid, coarse-grained directional guide:

$$
\mathbf{d}_{\text{real}} \approx \mathbb{E}[x_0 \mid x_{t_\phi}] - x,
\tag{23}
$$

which pulls the sample back towards the high-density region, bypassing the gradient vacuum encountered at $t_G$. $\qquad\square$

**MagicDistillation: Manifold Bridging via Energy Landscape Adaptation.** MagicDistillation (Shao et al., 2025) fine-tunes the real teacher $\phi$ on the student's generated samples before using it for distillation. Let $\phi_{\text{adapt}}$ denote the adapted teacher parameters.

**Proposition C.2.** *Adapting the teacher on student samples modifies the energy landscape to create a continuous gradient path ("bridge") connecting the student distribution to the target data distribution.*

*Proof.* Consider the standard real teacher $\phi_{\text{fix}}$ trained solely on $p_{\text{data}}$. Its energy potential $E_{\text{real}}(x) \approx -\log p_{\text{data}}(x)$ is undefined or essentially flat for samples in the Forbidden Zone $\mathcal{Z}_f$, where $p_{\text{data}}(x) \approx 0$. This results in a gradient $\nabla_x E_{\text{real}}(x) \approx \mathbf{0}$ or random noise.

MagicDistillation updates the teacher to minimize the denoising error on student samples $x \sim p_{\text{student}}$. This process effectively trains the teacher on a mixture distribution $p_{\text{mix}} = (1 - \lambda)p_{\text{data}} + \lambda p_{\text{student}}$. The adapted energy landscape can be approximated as:

$$E_{\text{adapt}}(x) \approx -\log\left((1 - \lambda)p_{\text{data}}(x) + \lambda p_{\text{student}}(x)\right). \tag{24}$$

For a distorted sample $x \in \mathcal{Z}_f$ where $p_{\text{data}}(x) \to 0$ but $p_{\text{student}}(x)$ is high, the adapted energy is dominated by the student term, ensuring the potential is well-defined. Crucially, since the adaptation is typically constrained (e.g., via LoRA or short-term finetuning), the teacher retains the global topology of $p_{\text{data}}$.

Therefore, the optimization creates a "bridge" in the energy landscape: $\nabla_x E_{\text{adapt}}(x)$ acts as a valid vector field that attracts $x$ from the student's current mode towards the target manifold. This eliminates the gradient discontinuity in $\mathcal{Z}_f$, allowing the student to traverse from $p_{\text{student}}$ to $p_{\text{data}}$ smoothly. $\qquad\square$

**DMD, DMD2, and DMDR: Deadlock Breaking via External Forces.** Unlike D-DMD and MagicDistillation which attempt to repair the internal gradient fields of the teachers, DMD, DMD2, and DMDR introduce an auxiliary external force $\mathbf{F}_{\text{ext}}$ to dominate the optimization when the primary distillation gradients vanish. We unify these methods under the *External Force Hypothesis*.

**Proposition C.3** (External Force Domination). *Let $\mathbf{g}_{DMD} = \nabla_x \mathcal{V}_{DMD}$ be the intrinsic distillation gradient. In the Forbidden Zone $\mathcal{Z}_f$, where $\|\mathbf{g}_{DMD}\| \to 0$ or becomes stochastic noise, the optimization trajectory is governed by the external auxiliary field $\mathbf{F}_{ext}$ provided by a third-party supervisor (Ground-truth, Discriminator, or Reward Model):*

$$x_{new} \leftarrow x - \eta(\underbrace{\mathbf{g}_{DMD}}_{\approx 0} + \lambda\mathbf{F}_{ext}) \approx x - \eta\lambda\mathbf{F}_{ext}. \tag{25}$$

We clarify the specific form of $\mathbf{F}_{\text{ext}}$ for each method below:

**DMD: Static Tethering via Regression.** Standard DMD (Yin et al., 2024b) employs an L2 regression loss on a small set of paired data $\{(z, x_{gt})\}$. The effective force is:

$$\mathbf{F}_{\text{DMD}} = \nabla_x \|x - x_{gt}\|^2 \propto (x - x_{gt}). \tag{26}$$

*Proof Sketch:* This force acts as a global linear spring (Hooke's Law). Regardless of whether $x \in \mathcal{Z}_f$ or how distorted the energy landscape $E_{\text{real}}(x)$ is, the vector $(x - x_{gt})$ always points strictly towards the ground truth sample. This "tethering" provides a robust, albeit rigid, recovery signal that drags the student out of the Forbidden Zone explicitly.

**DMD2: Adversarial Boundary Enforcement.** DMD2 (Yin et al., 2024a) removes the regression loss but introduces a GAN discriminator $D_\phi$ trained to distinguish real images from student samples. The external force is:

$$\mathbf{F}_{\text{DMD2}} = \nabla_x(-\log D_\phi(x)) = -\frac{\nabla_x D_\phi(x)}{D_\phi(x)}. \tag{27}$$

*Proof Sketch:* The discriminator $D_\phi$ is dynamically trained on the union of real data and current student samples (including those in $\mathcal{Z}_f$). Therefore, unlike the frozen Real Teacher which has no support in $\mathcal{Z}_f$, the discriminator explicitly learns the boundary of $\mathcal{Z}_f$. Specifically, $D(x)$ assigns low scores to samples in $\mathcal{Z}_f$, creating a steep gradient field $\nabla_x D(x)$ that pushes samples perpendicular to the decision boundary, effectively repelling them back towards the data manifold.

**DMDR: Asynchronous Reward Steering.** DMDR (Jiang et al., 2025) integrates Reinforcement Learning, using a reward model $R(x)$ to guide generation. The update can be viewed as an external force:

$$\mathbf{F}_{\text{DMDR}} \approx \nabla_x R(x). \tag{28}$$

*Proof Sketch:* We assume the reward model $R(x)$ (e.g., ImageReward or HPS) generalizes better than the generative density model due to its discriminative pre-training nature. In $\mathcal{Z}_f$, while the generative score $\nabla \log p(x)$ oscillates, the reward gradient $\nabla R(x)$ maintains a consistent direction towards high-fidelity regions (Assumption 3.3). This acts as an asynchronous steering wheel, overriding the confused signals from the teacher to navigate the student towards the region of interest defined by human preference.

# D. Limitation and Future Work

While AMD provides a unified and effective framework for multiphysics-aware distillation, it possesses certain limitations that open avenues for future research.

First, the efficacy of our self-correcting mechanism is inherently tied to the accuracy of **Forbidden Zone** ($\mathcal{Z}_f$) identification. Since AMD relies on a reward model to provide the advantage signal $\tilde{a}$, its performance is sensitive to the proxy's quality. If the reward model provides noisy or poorly calibrated feedback, it may fail to precisely detect $\mathcal{Z}_f$ pathologies, thereby preventing the adaptive mechanism from being fully triggered and leaving certain corrupted regions unrectified. Future work could explore more robust, unsupervised, or ensemble-based methods for $\mathcal{Z}_f$ detection to reduce dependency on a single external proxy.

Second, although our current adaptive operator $\mathcal{H}_{AMD}$ demonstrates significant performance gains, there remains vast room for more sophisticated adaptation strategies. Beyond the current formulation, future research could investigate integrating advanced optimization principles—such as the Momentum Trick, Orthogonal Gradient techniques, or Second-order information—into the teacher-student interplay. Such refinements could further stabilize the training trajectories and accelerate the student's convergence toward the true data manifold, particularly in high-dimensional and complex generative tasks.

# E. Algorithm

---

**Algorithm 1** Adaptive Distribution Matching Distillation (AMD)

---

**Input:** Student $G_\theta$, Real Teacher $\phi$, Fake Teacher $\psi$, Reward Model $\mathcal{R}$, Sensitivity $s$, Selection factor $k$, Group size $K$
**Output:** Optimized student parameters $\theta$

1 **while** *not converged* **do**
2     Sample a prompt $y \sim p_{\text{data}}$
3     Generate a group of $K$ samples: $\{x_i\}_{i=1}^{K}$ where $x_i = G_\theta(z_i, y)$
     `// Reward-aware Diagnosis`
4     $r_i = \mathcal{R}(x_i)$ for $i = 1 \ldots K$
5     $\mu_g = \text{mean}(r), \sigma_g = \text{std}(r)$
6     $\tilde{a}_i = \text{clip}((r_i - \mu_g)/(\sigma_g + \epsilon), -1, 1)$
7     Initialize gradients $\nabla_\theta \mathcal{L} \leftarrow 0, \nabla_\psi \mathcal{L}_\psi \leftarrow 0$
8     Sample timestep $t \sim \mathcal{U}[0, 1]$
9     **for** *each sample $x_i$ in group* **do**
10       Sample noise $\epsilon_i \sim \mathcal{N}(0, I)$
11       $x_{t,i} = \alpha_t x_i + \sigma_t \epsilon_i$
       `// 1. Student Update`
12       Get targets $\hat{x}_{0,\text{real}}$ from $\phi$ and $\hat{x}_{0,\text{fake}}$ from $\psi$
13       $\mathbf{d}_{\text{real},i} = x_i - \hat{x}_{0,\text{real}}, \quad \mathbf{d}_{\text{fake},i} = x_i - \hat{x}_{0,\text{fake}}$
14       $\alpha_i = 1 + s \cdot \tilde{a}_i, \quad \beta_i = 1 - s \cdot \tilde{a}_i$
15       $\mathbf{g}_i = \mathcal{H}_{AMD}\big(\mathbf{d}_{\text{real}}(x, \mathcal{F}_t, \phi), \mathbf{d}_{\text{fake}}(x, \mathcal{F}_t, \psi)\big)$ ;           `// Eq. 6 with score adaptation`
16       $\nabla_\theta \mathcal{L} \leftarrow \nabla_\theta \mathcal{L} + \mathbf{g}_i \frac{\partial G_\theta}{\partial \theta}$
       `// 2. Fake Teacher Update`
17       $w_i = \exp(-\tilde{a}_i)$ ;                     `// Higher weight for low rewards`
18       $\mathcal{L}_{\text{diff},i} = w_i \cdot \|\epsilon_i - \epsilon_\psi(x_{t,i}, t)\|^2$
19       $\nabla_\psi \mathcal{L}_\psi \leftarrow \nabla_\psi \mathcal{L}_\psi + \nabla_\psi(w_i \cdot \mathcal{L}_{\text{diff},i})$
20     **end**
21     Update student: $\theta \leftarrow \theta - \eta_\theta \nabla_\theta \mathcal{L}$
22     Update fake teacher: $\psi \leftarrow \psi - \eta_\psi \nabla_\psi \mathcal{L}_\psi$
23 **end**

---

# F. Supplementary Experimental Results

In this section, we present additional analytical experiments, quantitative evaluations, and qualitative demonstrations to further validate the effectiveness of AMD.

## F.1. Analytical Experiments

In this section, we provide empirical evidence to support the design motivation of AMD.

**Navie Adaption vs AMD.** According to our unified optimization framework in Equation (6), the choice of the adaptation operator $\mathcal{H}$ fundamentally dictates the distillation dynamics. To investigate this, we compare a **naive adaptive strategy** ($\mathcal{H}_{\text{naive}}$, see Equation (8)) with our proposed **decoupled modulation** ($\mathcal{H}_{\text{AMD}}$, see Equation (10)) on a 2D multi-modal toy dataset. In this setup, the real teacher's energy potential represents a symmetric multi-modal distribution, while the reward model favors a specific subset of modes (located in the lower-left region).

As illustrated in Fig. 8, although both methods are guided by the same reward proxy, their training stability and convergence behavior differ significantly:

- **Naive Adaptation:** Simple linear scaling of $\mathbf{d}_{\text{real}}$ and $\mathbf{d}_{\text{fake}}$ fails to resolve the structural conflict between the teacher's global guidance and the reward model's local prioritization. In the mid-to-late stages of training (Iteration 1000–2000), this leads to a **catastrophic distribution collapse**, where the student distribution loses its multi-modal structure and dissolves into an uninformative mass.

- **AMD (Ours):** By contrast, AMD successfully navigates the student toward the high-reward regime while maintaining the structural integrity of the generation manifold. By decoupling the signals into distribution matching (DM) and conditional alignment (CA) terms, AMD enables stable, high-fidelity steering that avoids the "gradient deadlock" encountered by the naive approach.

These results empirically validate the necessity of the decoupled adaptation mechanism in AMD for achieving stable, reward-aware distribution matching.

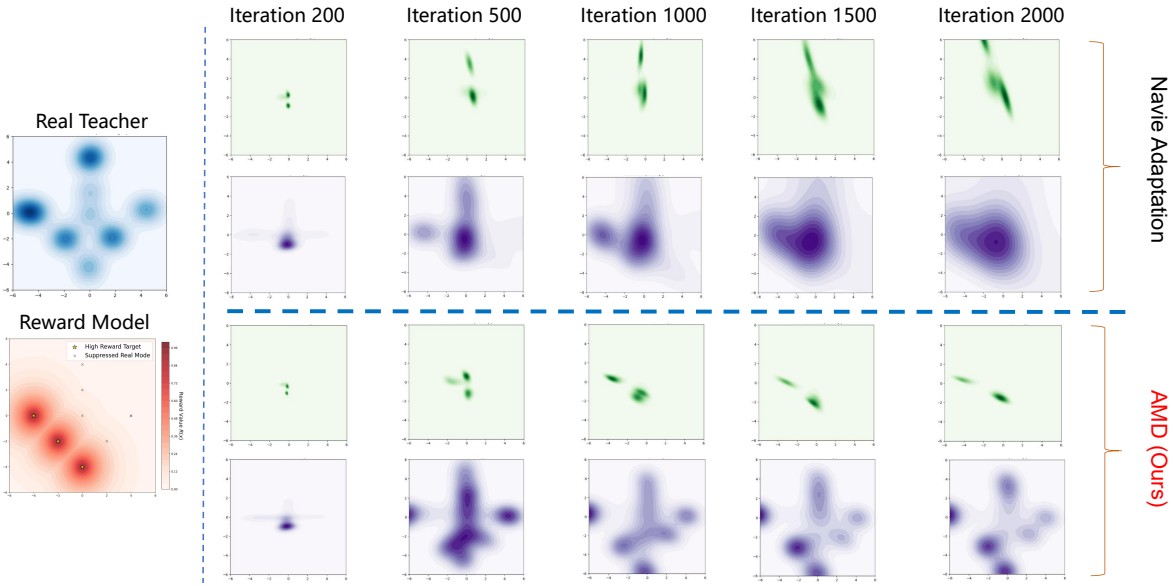

*Figure 8.* We track the evolution of the student distribution on a 2D toy dataset. **Top:** Naive Adaptation ($\mathcal{H}_{\text{naive}}$) suffers from mode merging and eventual distribution collapse as it fails to balance the competing distillation forces. **Bottom:** AMD ($\mathcal{H}_{\text{AMD}}$) successfully maneuvers the student toward high-reward modes (lower-left) while preserving sharp, distinct distributional fidelity. This highlights the criticality of decoupled signal modulation in preventing training instability within the *Forbidden Zone*.

## F.2. Quantitative Results

**Results of AMD in other Image Generation Benchmark** Here, we first provide the breakdown of the experiments on GenEval, as shown in Tab. 8, our proposed AMD almost outperforms the previous methods across all dimensions. Besides, we conduct the experiments on DrawBench, as shown in Tab. 9, the experiment results further validate the effectiveness of our method. Last but not least, we provide the winning rate of AMD across different datasets, compared with the standard DMD2.

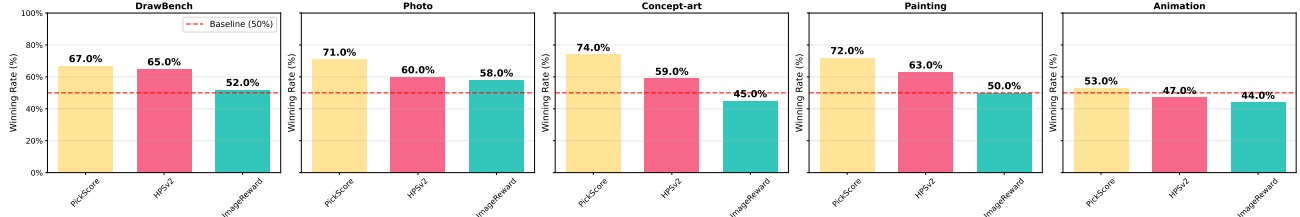

*Figure 9.* The winning rate of AMD over DMD2 on SDXL across DrawBench and HPDv2. The standard DMD2 (baseline) winning rate defaults to 50%. The results reveal the superiority of AMD in synthesizing images with good quality, comparing with DMD2.

*Table 8.* Comparison of different text-to-image models on GenEval (Ghosh et al., 2023) benchmark. The best results in each category are highlighted in **bold**.

| Method | #Params | Resolution | NFEs | Overall ↑ | Single ↑ | Two ↑ | Count ↑ | Colors ↑ | Pos ↑ | Attr. ↑ |
|---|---|---|---|---|---|---|---|---|---|---|
| *Pretrained Models* | | | | | | | | | | |
| SDXL (Podell et al., 2023) | 2.6B | $1024 \times 1024$ | $50 \times 2$ | 0.55 | 0.98 | 0.74 | 0.39 | 0.85 | 0.15 | 0.23 |
| FLUX.1-dev (Labs, 2024) | 12.0B | $1024 \times 1024$ | 50 | 0.66 | 0.98 | 0.81 | 0.74 | 0.79 | 0.22 | 0.45 |
| *Distilled Models* | | | | | | | | | | |
| SDXL-LCM (Luo et al., 2023) | 2.6B | $1024 \times 1024$ | 4 | 0.50 | 0.99 | 0.55 | 0.38 | **0.85** | 0.07 | 0.14 |
| SDXL-Turbo (Podell et al., 2023) | 2.6B | $512 \times 512$ | 4 | 0.56 | **1.00** | 0.72 | **0.49** | 0.82 | **0.11** | 0.21 |
| SDXL-Lightning (Lin et al., 2024) | 2.6B | $1024 \times 1024$ | 4 | 0.53 | 0.98 | 0.61 | 0.44 | 0.84 | **0.11** | 0.21 |
| SDXL-DMD2 (Yin et al., 2024a) | 2.6B | $1024 \times 1024$ | 4 | 0.51 | 0.98 | 0.62 | 0.43 | 0.82 | 0.07 | 0.15 |
| SDXL-DMDR (Jiang et al., 2025) | 2.6B | $1024 \times 1024$ | 4 | 0.56 | 0.99 | 0.76 | 0.42 | 0.84 | **0.11** | **0.24** |
| SDXL-AMD (Ours) | 2.6B | $1024 \times 1024$ | 4 | **0.57** | **1.00** | **0.76** | 0.47 | **0.85** | 0.10 | 0.23 |

*Table 9.* Quantitative comparison of text-to-image task on DrawBench (Rombach et al., 2022) dataset. (base model: SDXL). We compare AMD against DMD2 using a lot of preference-based metrics. The results validate the effectiveness of AMD.

| Method | PickScore | HPSv2 | ImageReward |
|---|---|---|---|
| DMD2 | 22.19 | 29.66 | 79.46 |
| AMD (Ours) | **22.49** | **30.40** | **84.01** |

**Results of AMD in other Video Generation Benchmark** We further evaluate the scalability of AMD by applying it to the larger Wan2.1-14B backbone. As shown in Tab. 10, AMD achieves a higher overall performance than DMD2 on our internal benchmark, improving the Total Score from 118.61 to **122.15**.

Our internal benchmark consists of 419 diverse image prompts collected from the web, covering a wide range of visual styles (e.g., anime, human-centric scenes, and realistic physical environments), as well as both landscape and portrait layouts. For each image, captions are generated using Gemini 2.5 Pro to ensure rich and consistent textual descriptions. We plan to release this benchmark publicly in future work.

Notably, we observe a trade-off between Visual Quality (VQ) and Text Alignment (TA), which is consistent with our findings on the Wan2.1-1.3B model. This behavior can be attributed to the VideoAlign reward, which explicitly encourages dynamic motion generation, leading to a substantial improvement in Motion Quality (MQ, +16.24), sometimes at the expense of static visual fidelity and strict text-image alignment. Furthermore, as reported in Tab. 11, evaluation under the VBench++ (Huang et al., 2025c) framework shows that AMD consistently outperforms DMD2.

*Table 10.* Quantitative comparison of Image-to-Video generation task. (base model: Wan2.1-14B). Comparison between AMD and DMD2 across VBench and Internal metrics.

| Method | VBench-I2V | | | | Internal Bench | | | |
|---|---|---|---|---|---|---|---|---|
| | VQ ↑ | MQ ↑ | TA ↑ | Total ↑ | VQ ↑ | MQ ↑ | TA ↑ | Total ↑ |
| DMD2 | 26.48 | 32.62 | **67.26** | 126.36 | **12.90** | 5.27 | **100.47** | 118.61 |
| AMD (Ours) | **30.56** | **39.49** | 60.67 | **130.72** | 11.26 | **21.51** | 89.30 | **122.15** |

*Table 11.* Quantitative comparison of Image-to-Video generation task. (base model: Wan2.1-14B). Best results are highlighted.

| Method | I2V Subject (↑) | Subject Consistency (↑) | Motion Smoothness (↑) | Aesthetic Quality (↑) | Imaging Quality (↑) |
|---|---|---|---|---|---|
| ***100-NFE*** | | | | | |
| CogVideoX-I2V-SAT (Yang et al., 2024) | 97.67 | 95.47 | 98.35 | 59.76 | 67.64 |
| I2Vgen-XL (Zhang et al., 2023) | 97.52 | 96.36 | 98.31 | 65.33 | 69.85 |
| SEINE-512x320 (Chen et al., 2023) | 96.57 | 94.20 | 96.68 | 58.42 | 70.97 |
| VideoCrafter-I2V (Chen et al., 2024) | 91.17 | 97.86 | 98.00 | 60.78 | 71.68 |
| SVD-XT-1.0 (Blattmann et al., 2023) | 97.52 | 95.52 | 98.09 | 60.15 | 69.80 |
| Step-Video-TI2V (Ma et al., 2025) | 95.50 | 96.02 | 99.24 | 70.44 | 78.82 |
| ***4-NFE*** | | | | | |
| DMD2 (Yin et al., 2024a) | 0.9612 | 0.9671 | 0.9908 | 0.6668 | 0.7092 |
| AMD (Ours) | **0.9663** | **0.9843** | **0.9935** | **0.6705** | **0.7098** |

## F.3. Visualization Results

We provide an extensive gallery of visual samples generated by AMD across both image and video generation tasks, highlighting its superiority in fidelity and semantic alignment compared to standard distillation methods

**Image Generation**    We present a broader range of image generation results across different datasets, including DrawBench and HPDv2, using AMD distilled from SDXL. As shown in Fig. 10, 11, 12, 13, and 14, the visual results demonstrate the effectiveness of our proposed AMD, compared with standard DMD2, where synthesized image quality and texture have greatly improved.

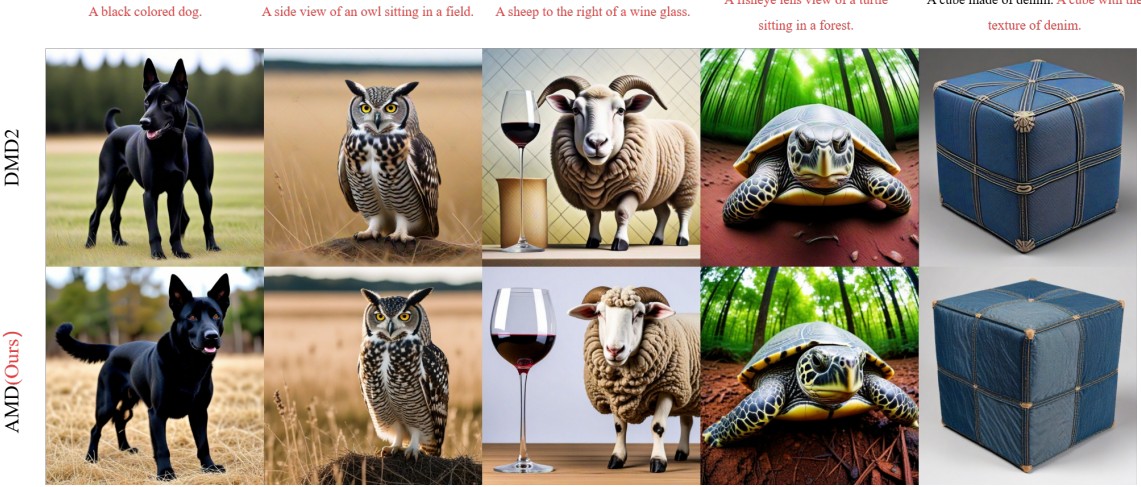

*Figure 10.* Synthesized images of ADM distilled from SDXL on DrawBench.

A portrait of a character in a scenic environment.

The image is a stunning illustration of a knight warrior wearing Nordic armor and a Skyrim mask, with intricate details and dynamic lighting that make it perfect for RPG portraits and cosplay.

An ancient statue of a mushroom goddess wearing pagan clothes and leaves, located in a cedar forest.

Pippi is tethered to the international space station in her space suit amidst stars and galaxies.

A 3D render of a cyberpunk-necromancer with intricate details and believable eyes, depicted in a front-facing, symmetrical view as epic fantasy art.

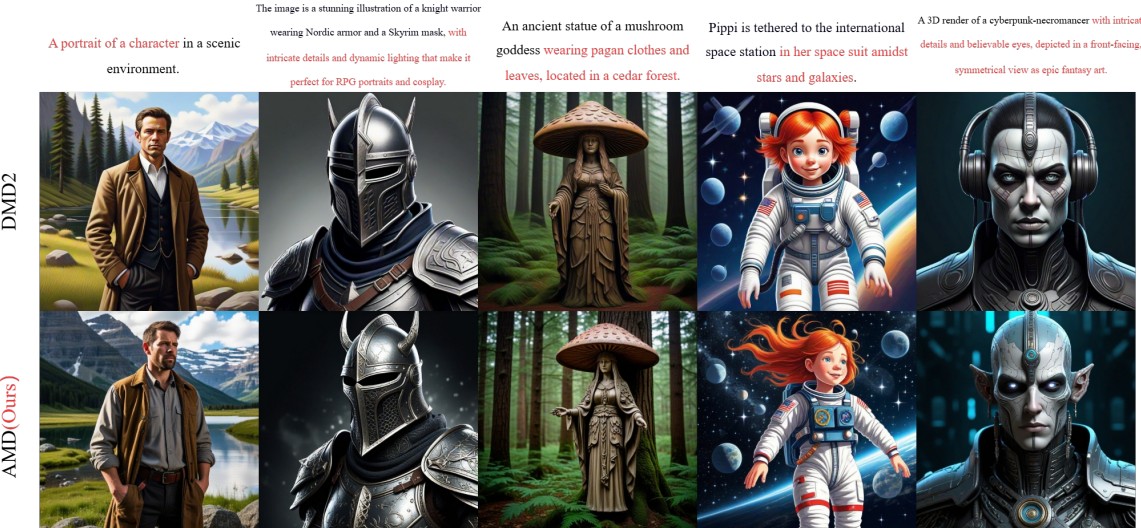

*Figure 11.* Synthesized images of ADM distilled from SDXL on concept-art subset of HPD v2.

A dilapidated shack hidden in a misty, overgrown Witchwood forest inhabited by evil fairies, depicted in a detailed, ink illustration by Greg Rutkowski.

A surreal portrait of a woman with a giant carnation face in a flower field at sunset with colorful clouds and a large sky, created by artist Simon Stålenhag

Museum painting of a mouse stealing cheese artwork.

A monkey in a blue top hat painted in oil by Vincent van Gogh in the 1800s.

Colorful illustration of a forest tunnel illuminated by sunlight and filled with wildflowers.

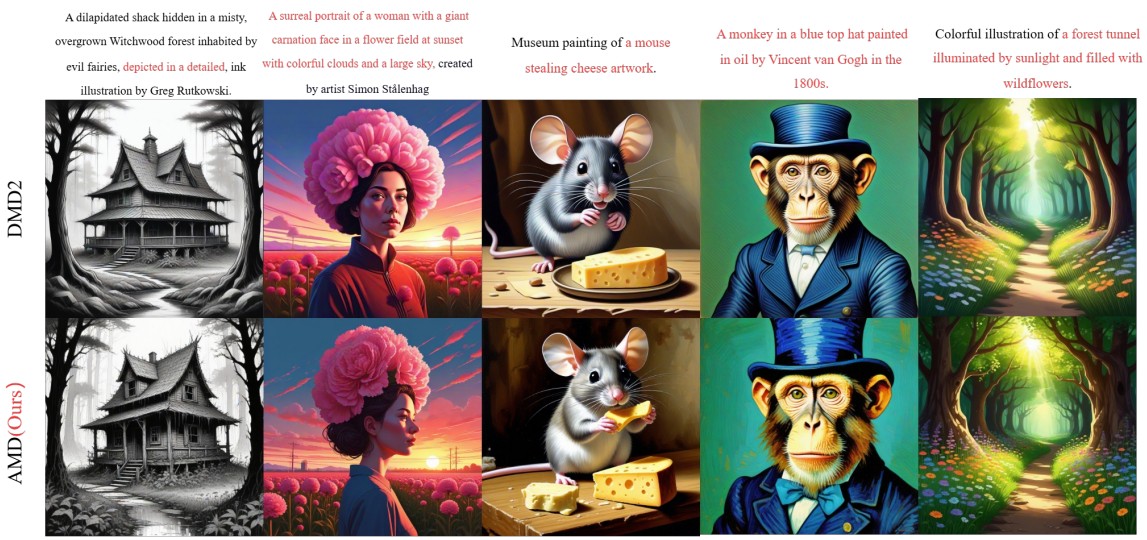

*Figure 12.* Synthesized images of ADM distilled from SDXL on painting subset of HPD v2.

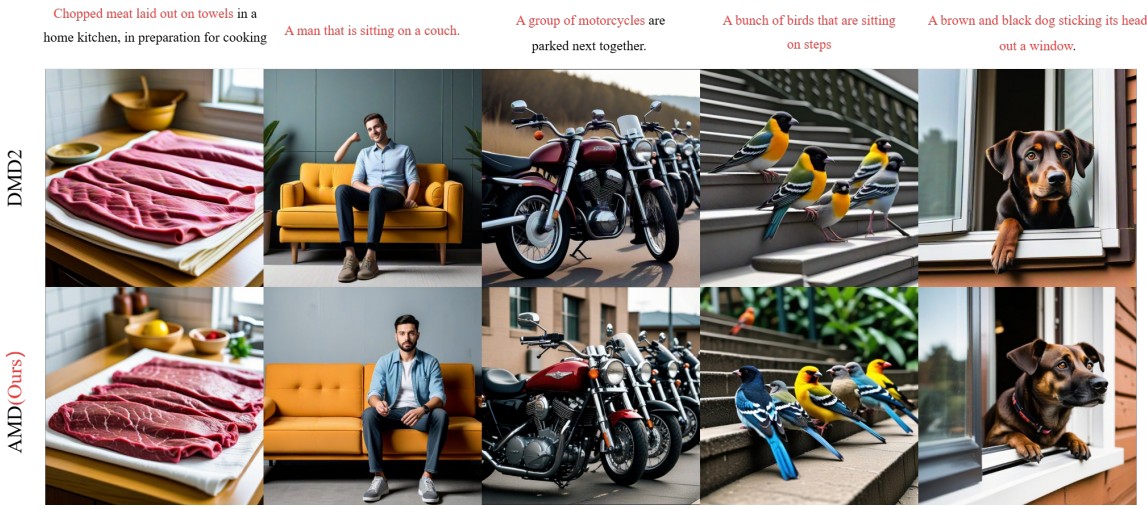

*Figure 13.* Synthesized images of ADM distilled from SDXL on photo subset of HPD v2.

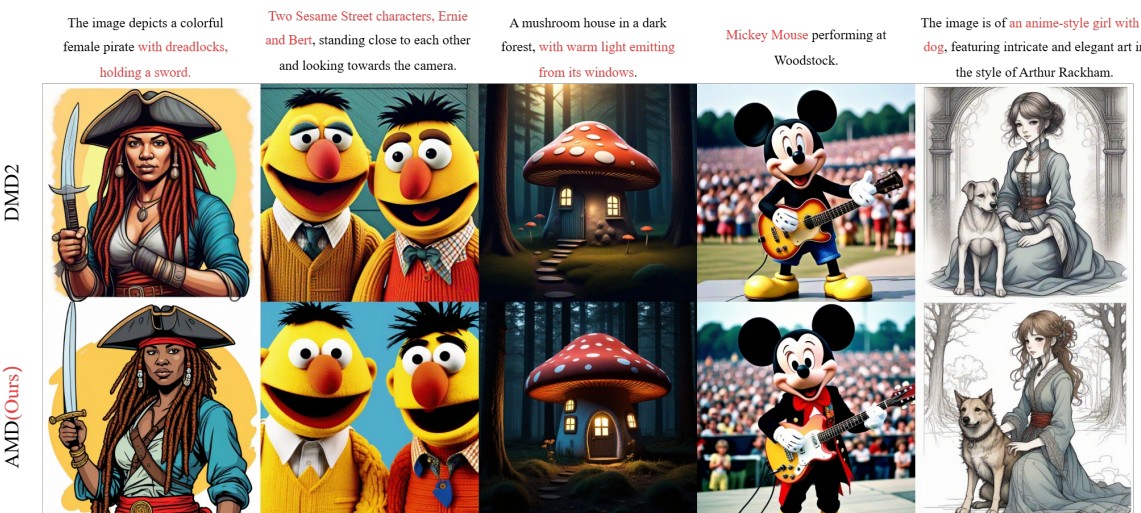

*Figure 14.* Synthesized images of ADM distilled from SDXL on anime subset of HPD v2.

**Video Generation** We present a broader range of video generation results using the Wan2.1-1.3B backbone. Fig. 15 visually compares videos distilled via standard DMD versus our AMD. It is evident that the baseline DMD often suffers from temporal flickering or motion degradation (e.g., static backgrounds or unnatural warping) in challenging scenarios. In contrast, AMD preserves dynamic fidelity and generates smoother, more logically consistent motion, demonstrating the efficacy of our decoupled modulation strategy in the spatiotemporal domain.

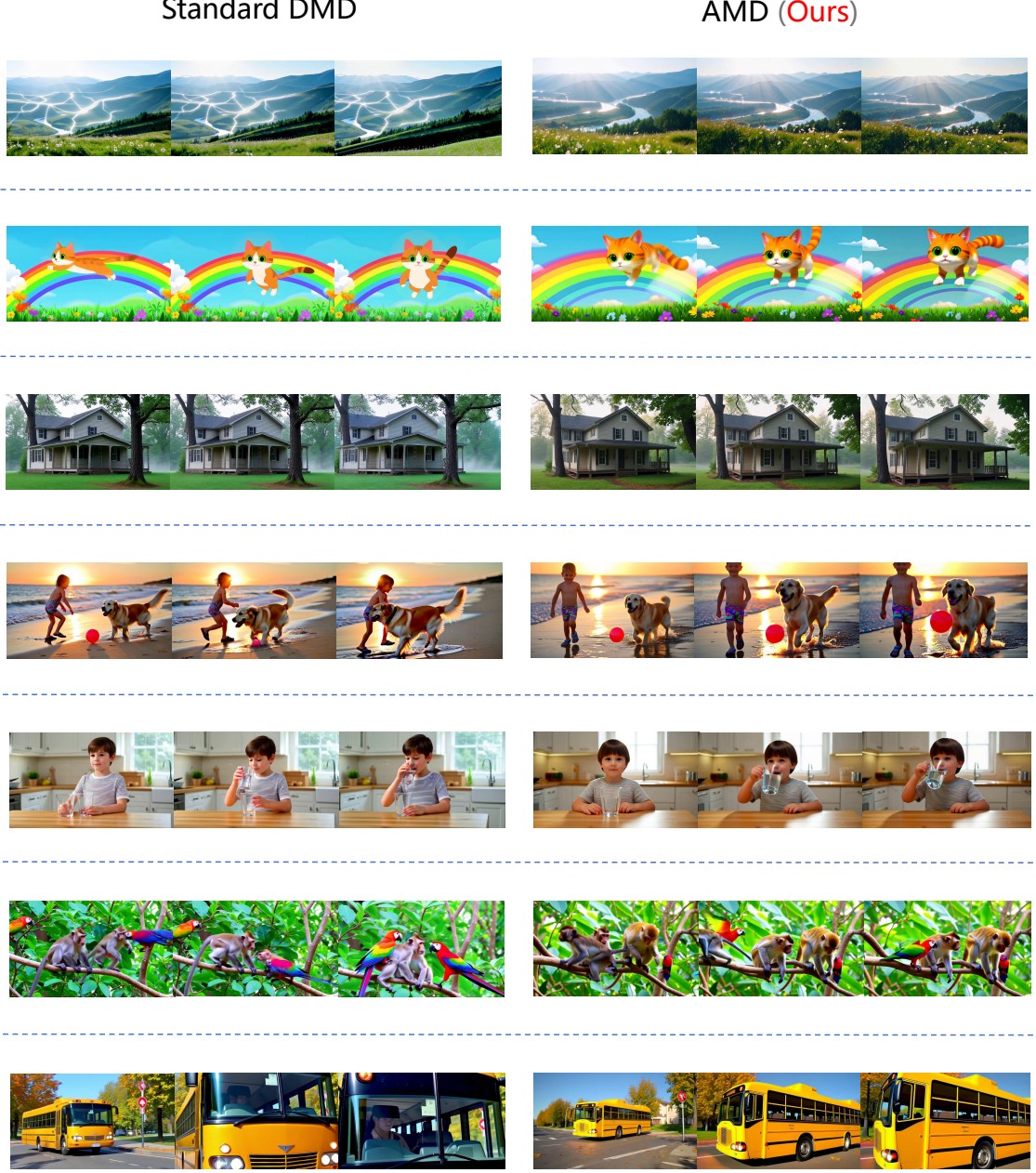

*Figure 15.* Qualitative comparison on text-to-video generation. We show that AMD significantly outperforms the baseline, exhibiting superior motion smoothness, higher visual fidelity, and better prompt alignment.

