# OpenReview forum: "Optimizing Few-Step Generation with Adaptive Matching Distillation"
_ICML.cc/2026/Conference — ICML 2026 regular_

### Official Review · Reviewer_9N9X · 2026-03-11

**Soundness:** 2
**Presentation:** 3
**Significance:** 3
**Originality:** 2
**Overall Recommendation:** 4
**Confidence:** 4

**Summary:**

The paper finds a Forbidden Zone in DMD. This zone occurs when the teacher gives unreliable scores. The fake score simultaneously provides weak repulsion in this region. The authors reformulate DMD as a latent optimization problem. They propose Adaptive Matching Distillation (AMD). AMD uses a reward model as a proxy. It detects these corrupted zones during training. The method dynamically adjusts the gradient components based on this reward. It also sharpens the fake teacher's repulsive landscape. Experiments cover both image and video generation. The authors evaluate on SDXL, SiT, and Wan2.1. Results show consistent improvements over existing DMD baselines.

**Compliance With Llm Reviewing Policy:**

Affirmed.

**Final Justification:**

The authors' responses resolved my concerns, which proves the effectiveness of AMD. Therefore, I am willing to raise my score to Weak Accept.

**Key Questions For Authors:**

1. What is the exact wall-clock training overhead? Please quantify the cost of generating $K$ samples and computing rewards at every training step (Also compare with DMD/DMD2).
2. What is the group size used in the experiments?
3. Did the experiments use LoRA or full parameter training?
4. For Image Generation task, could you please show the training results that initialize the student from the teacher (i.e. do not use  two-stage training protocol)?
5. How sensitive is the AMD to the choice of the reward model?
6. Does the exponential weighting in Eq. 14 risk gradient explosions?
7. How do you stabilize the fake teacher training when sample advantages are somehow skewed?
8. Minor: L750 missing table reference.

**Limitations:**

The authors briefly mention the dependency on reward model quality in the impact statement. However, they do not adequately discuss the computational limitations. The method clearly introduces massive training overhead via the group generation step. The authors must explicitly discuss this trade-off. They should also discuss the potential for reward hacking in greater detail.

**Strengths And Weaknesses:**

- **Soundness** The proposed AMD method is highly intuitive. However, Assumption 3.3 is very strong. Reward models are notoriously prone to hacking. The paper assumes reward perfectly correlates with teacher competence. This is not always true. Furthermore, the computational cost is ignored. The algorithm requires generating $K$ samples per step. It also requires $K$ forward passes through a reward model. This adds massive overhead.

- **Presentation** The paper is generally well-structured. The diagrams are highly informative. The unified perspective in Section 3.1 is good. The derivation in Appendix C.1 is slightly loose. It jumps between parameter space and latent space updates. A more rigorous mathematical notation would separate these spaces clearly.

- **Significance** Accelerating diffusion models is a highly relevant problem. DMD is a leading paradigm in this space. Improving its training stability has great practical value.

- **Originality** Reinterpreting DMD variants as gradient corrections is a fresh perspective. The specific signal decomposition in Eq 10 is original. Reweighting the fake teacher's loss based on sample advantage is also a neat contribution.

---

> ### Author Rebuttal · Authors · 2026-03-31
>
> We sincerely thank the reviewer for the detailed and constructive feedback. Below we clarify the main concerns regarding the reward-model assumption, computational overhead, and implementation details.
>
> ---
>
> **W1**: Reward models are notoriously prone to hacking. Assumption 3.3 is not always true.
>
> **A**: We thank the reviewer for recognizing the intuition behind AMD. We want to reiterate that  the reward model in  AMD serves as a practical proxy for forbidden-zone detection, not an oracle. Under our unified view, AMD only requires a reasonably informative indicator rather than a perfectly calibrated one. The choice of a reward model is an engineering trade-off for performance and speed, it is possible to not rely on an external reward model but only on an internal proxy.
>
> A more intuitive view of this visualization is available at the anonymous link <https://anonymous.4open.science/api/repo/anonymous-ICML-Submission-10939-CB92/file/index.html?v=7dabd7eb>. (Due to space constraints, please refer to our response to Reviewer ioBW’s **Q2**.)
>
> ---
>
> **W2**:  The computational cost is ignored, and AMD adds massive overhead.
>
> **A**: Thank you for your question. The additional computational cost is minimal, as we perform sampling and reward evaluation only once every $N$ steps to balance efficiency and performance. For instance, in our SDXL experiments, this is done every 8 steps, making AMD only moderately slower than DMD while still achieving clear performance gains—an overhead that is already relatively small compared to DMDR. We will clarify this in the revision.
>
> ---
>
> **W3**: The derivation in Appendix C.1 is slightly loose.
>
> **A**: We thank the reviewer for the positive assessment of the paper’s presentation and structure. As noted in the original submission, this appendix is intended to supplement the visualization and intuition in Fig. 2, rather than to serve as the primary formal derivation.  We will refine the derivation in Appendix C.1 for greater clarity.
>
> ---
>
> **Q1**: What is the exact wall-clock training overhead?
>
> **A**: Please see **W2**.
>
> ---
>
> **Q2**: What is the group size used in the experiments?
>
> **A**: For video generation models (e.g., Wan 1.3B/14B), we use a group size of 8, while for image generation models (e.g., SDXL), we use a group size of 64. This choice reflects the different computational budgets of video and image training. These details will be added in the revision.
>
> ---
>
> **Q3**: Did the experiments use LoRA or full parameter training?
>
> **A**: To ensure a fair comparison with prior DMD-style methods, we follow the official default setting of DMD2 and use full-parameter training. We will clarify this implementation detail in the revised version.
>
> ---
>
> **Q4**: For image generation, could you show results with student initialized from teacher (i.e., no two-stage training)?
>
> **A**: We adopt this comparison protocol to stay consistent with the official implementation of DMDR, whose SiT-XL/2 experiments use this two-stage training setup. For a fair comparison, we therefore follow their official implementation as closely as possible. In contrast, for SDXL, Wan 14B/1.3B, we also follow the standard DMD2 setting. We hope this clarifies the reviewer’s concern.
>
> ---
>
> **Q5**: How sensitive is the AMD to the reward model?
>
> **A**: AMD is not highly sensitive to the reward model itself. The reward is used only as a practical proxy for forbidden-zone detection, rather than as an oracle. This is further supported by our additional analyses: AMD remains effective under moderate Gaussian perturbations to the reward score (iobW’s **Q1**), and it also works with a teacher-intrinsic density proxy (ioBW’s **Q2**). More importantly, we would like to emphasize again that the main value of AMD lies in its unified perspective, while the reward model is only one practical way to instantiate forbidden-zone detection under a performance-efficiency trade-off.
>
> ---
>
> **Q6**: Does the exponential weighting in Eq. 14 risk gradient explosions?
>
> **A**: In practice, we did not observe gradient explosion caused by the exponential weighting in Eq. 14. As shown in **Q7**, clipping keeps the exponent well controlled and prevents extreme weights.
>
> ---
>
> **Q7**: How to stabilize the fake teacher training when sample advantages are somehow skewed?
>
> **A**: We clip the sample advantage $\tilde {a}_ i$ in Eq.12 to $ \[ -1, 1 \] $, which prevents extreme values caused by skewed reward distributions and stabilizes fake-teacher training. This also helps avoid gradient explosion or unstable updates, and is consistent with common stabilization practices such as GRPO.
>
> ---
>
> **Q8**: L750 missing table reference.
>
> **A**: Thank you for catching this. We will fix the missing table reference in the revised version.
>
> ---
>
> Finally, we sincerely thank the reviewer again for the detailed and constructive feedback. We believe these comments help further strengthen the paper, and we also look forward to any further discussion.

---

> > ### Author Rebuttal · Reviewer_9N9X · 2026-04-04
> >
> > Thanks for the authors' response. My concerns are resolved

---

> > > ### Author Response · Authors · 2026-04-04
> > >
> > > We sincerely thank you again for your hard work and helpful comments.
> > >
> > > We are very glad to see that your concerns are fully resolved.
> > >
> > > We also respectfully encourage the reviewer to consider adjusting the score accordingly, as the official instruction suggests. Your work is a great addition to the community.

---

### Official Review · Reviewer_iobW · 2026-03-11

**Soundness:** 3
**Presentation:** 3
**Significance:** 4
**Originality:** 4
**Overall Recommendation:** 5
**Confidence:** 4

**Summary:**

This paper analyzes the instability of Distribution Matching Distillation (DMD) caused by “Forbidden Zones” far from the data manifold and proposes Adaptive Matching Distillation (AMD), a unified and self-correcting framework with dynamic score adaptation and repulsive landscape sharpening to guide samples back to valid distributions and enable stable, high-fidelity few-step generation.

**Compliance With Llm Reviewing Policy:**

Affirmed.

**Final Justification:**

Thank you for the detailed rebuttal. My main concerns have been addressed, and I will maintain my score.

**Key Questions For Authors:**

1. If the authors can demonstrate AMD's robustness to noise in these proxy signals, it would significantly increase my score for Soundness and Significance.

2. The paper defines Forbidden Zones as regions where the real teacher's guidance is unreliable. Is there a formal way to quantify the boundaries of these zones without relying on external proxies like DINOv2? For instance, can the disagreement (variance) between the real and fake teacher scores be used as an intrinsic diagnostic tool instead of an external one?

**Limitations:**

Yes

**Strengths And Weaknesses:**

**Strengths**
1. Strong theoretical insight: Introduces a Sample-space Gradient Descent Reformulation that provides an intuitive explanation of DMD dynamics in latent space.

2. Clear presentation and practical relevance: Well-structured paper with effective visualizations addressing stability issues in few-step diffusion distillation.

3. Comprehensive experiments: Validated across multiple modalities and architectures with diverse evaluation metrics.


**Weaknesses**
1. The practical execution of AMD relies significantly on the quality of the pre-trained reward models used as "diagnostic sensors." A potential concern regarding soundness is that if these reward models themselves possess uncalibrated regions or their own "Forbidden Zones," the corrective signals provided to the student model could become misleading.

2. The experimental results in video generation reveal a slight trade-off.

---

> ### Author Rebuttal · Authors · 2026-03-31
>
> We sincerely thank the reviewer for the positive assessment of our work and for recognizing the theoretical insight, unified perspective, and comprehensive experiments. Below we address the main concerns.
>
> ---
>
> **W1**: Use of pretrained reward model as diagnostic sensors.
>
> **A**:  We agree that using a pretrained reward model introduces an additional practical component. However, we would like to emphasize that our intention is not to treat the reward model as an additional teacher, but rather as a convenient proxy for identifying samples that are likely to lie in forbidden zones. In this sense, AMD fundamentally relies on a forbidden-zone indicator, rather than on any specific external reward model.
>
> According to the reviewers’ suggestions, we further discuss in **Q2** that this proxy can also be instantiated by teacher-intrinsic signals, which highlights the generality of the AMD framework.
>
> ---
>
> **W2**: Slight trade-off in video generation.
>
> **A**: Yes, we agree that there is a slight trade-off in the video setting. As noted in **W1**, when AMD instantiates forbidden-zone detection with a reward model, it can inherit some limitations commonly associated with reward-based methods. In particular, improving video quality may come with a slight decrease in prompt alignment. We will clarify this trade-off more explicitly in the revised version. Importantly, we view this as a property of the specific proxy choice rather than of the AMD framework itself; we further discuss this point in **Q2**, where we show that AMD is not inherently tied to any particular proxy.
>
> ---
>
> **Q1**: Robustness to noisy proxy signals.
>
> **A**: We agree that robustness to imperfect proxy signals is important. To directly evaluate this, we conducted an additional ablation by perturbing the reward proxy with Gaussian noise, the results are as follows:
>
> | $\sigma$ | FID | sFID | IS |
> |---|---:|---:|---:|
> | DMD | 3.5573 | 5.8499 | 314.42 |
> | AMD w/o perturbation | **3.4690** | 5.7464 | **316.02** |
> | AMD w/ perturbation | 3.7153 | **5.2638** | 315.92 |
>
> This suggests that moderate proxy noise acts as a form of 'regularization': it suppresses overly sharp guidance from the proxy signal and encourages the generator to explore more diverse spatial layouts while preserving class fidelity, thereby significantly improving sFID but slightly degrading FID.
>
> ---
>
> **Q2**: Can forbidden zones be quantified intrinsically, e.g., via teacher disagreement?
>
> **A**:  Yes. Forbidden zones refer to regions that are insufficiently supported by the underlying data distribution, and according to Definition 3.2, they can in principle be indicated by the magnitude of  $\log p(x)$.  Therefore, beyond reward-based proxies, we believe there are also intrinsic ways to quantify forbidden zones using statistics derived from the model itself.
>
> For example, in [1], $\log p(x)$ can be approximated by using an ELBO-style score as  $-\mathbb{E}_{t,\epsilon} \left[ \|| \epsilon - \epsilon _\theta (x_t) \||^2 \right] + C ,$
>
> which depends only on the teacher itself.  In addition, inspired by the reviewer’s suggestion, we also explored using the discrepancy between the real teacher and the fake teacher as an intrinsic proxy. Combining these two ideas, we conducted ImageNet experiments without introducing any external reward model and observed improvements, as shown below.
>
> | Method | FID | sFID | IS |
> |---|---:|---:|---:|
> | DMD | 3.5573 | 5.8499 | 314.42 |
> | AMD (external proxy) | **3.4690** | 5.7464 | **316.02** |
> | AMD (intrinsic proxy) | 3.6860 | **5.4793** | 315.04 |
>
> Here, we observe that the intrinsic proxy improves sFID and IS, while slightly worsening FID. This is a reasonable outcome: the intrinsic proxy tends to favor higher-density regions of the training distribution, thereby making training more stable, at the cost of some diversity. This is also consistent with prior observations on DMD-style methods.
>
> In particular, we further visualized this intrinsic reward in a 2D experiment (see **external link** <https://anonymous.4open.science/api/repo/anonymous-ICML-Submission-10939-CB92/file/index.html?v=7dabd7eb>). The visualization shows that the intrinsic reward covers the high-probability support regions of the teacher quite well. We hope this provides further evidence that intrinsic forbidden-zone diagnostics are both feasible and meaningful.
>
> More broadly, we hope this highlights a key value of AMD: it provides a unified view of distribution matching methods, in which forbidden-zone detection is one possible instantiation rather than a framework-specific requirement. In this sense, AMD is general and not tied to any particular proxy.
>
> ---
>
> Finally, we sincerely thank the reviewer again for the thoughtful and constructive feedback. We hope our discussion further clarifies this perspective and look forward to any further discussion.
>
> ---
>
> Reference
>
> [1] Li et al. (2023). Your diffusion model is secretly a zero-shot classifier. ICCV 2023.

---

> > ### Author Rebuttal · Reviewer_iobW · 2026-04-03
> >
> > All my concerns have been addressed.

---

> > > ### Author Response · Authors · 2026-04-07
> > >
> > > Thank you for your feedback and for the constructive review process. We sincerely appreciate your time, consideration, and valuable insights. We are glad that our clarifications addressed your concerns, and we truly appreciate your recognition of our work.

---

### Official Review · Reviewer_jg1g · 2026-03-12

**Soundness:** 2
**Presentation:** 3
**Significance:** 2
**Originality:** 2
**Overall Recommendation:** 4
**Confidence:** 3

**Summary:**

This paper proposes the Adaptive Matching Distillation (AMD) algorithm, which introduces reward-based weighting of the Decoupled DMD loss and fake score teacher. The authors argue that reward can serve as a proxy for identifying _forbidden zones_, or, put simply, low-density regions, where a real teacher doesn’t provide a meaningful distillation signal because samples are out of the training manifold. The fake score, on the other hand, doesn’t act as a strong repellent in these regions. Authors reinterpret previous DMD-based distillation methods according to this view and show that, unlike them, adaptive reward-based weighting of the DMD loss can achieve superior results without an additional external gradient force. Finally, authors conduct experiments on image/text-to-image/video generation benchmarks, on which their methods demonstrate competitive or better results than the baseline distillation methods.

**Compliance With Llm Reviewing Policy:**

Affirmed.

**Final Justification:**

All of the raised concerns are addressed clearly in the rebuttal. I appreciate the authors' clarification regarding the intended scope of the paper, the connection between forbidden zones and adaptive weighting, and experimental results in Table 2. However, I am still not entirely convinced by the theoretical grounding and significance of the unifying view on the DMD-based distillation methods that follows from Proposition 3.1. I appreciate that the authors will clarify the scope and approximations behind it in the revised version.
That said, I find that AMD is a useful and sufficiently well validated practical technique for distillation. Therefore, I am willing to raise my score to "Weak Accept".

**Key Questions For Authors:**

1. How could the proposed adaptive scheme be applied to unconditional generation? Are multi-modal 2D experiments conducted in an unconditional setting?
2. Could the authors elaborate on how reward-based weighting affects the distillation dynamics at different noise levels?

**Limitations:**

Yes

**Strengths And Weaknesses:**

## Soundness/Presentation
### Strengths
1. A large share of experiments indicate that the proposed adaptive distillation scheme results in good quality of distillation across a large number of benchmarks.
2. There are large-scale experiments on different domains, including video generation.
### Weaknesses
1. The adaptive weighting scheme makes sense. On the one hand, when the generator's outputs are far off the teacher's manifold, lower $\tilde{a}$ (or bigger $\beta(\tilde{a})$)  results in more emphasis on the DM term and, in turn, pushes the samples toward a valid manifold. On the other hand, if the generator already produces good images, then the $\alpha(\tilde{a})$ is bigger and the result is a stronger emphasis on the CA term, which enforces semanticity. From this, it seems to me that the DM term is largely responsible for pushing the model towards the teacher's higher density regions (i.e., escaping *forbidden zones*), and that additional reward weighting just accelerates this process. Having said that, in my view, the connection between the notion of *forbidden zones* and the adaptive weighting scheme is a bit vague, and the latter looks more like a convenient (working!) heuristic, which doesn't follow from the previous sections.
   I believe that the paper would benefit from a deeper theoretical and empirical investigation of the *forbidden zones* and their impact on DMD distillation dynamics at different noise levels.
2. The assumptions behind Proposition 3.1, which reduce the kernel $K_{\theta} = J_{\theta}J_{\theta}^T$ to an effective scalar $\eta_{\text{eff}}$ are not clear and seem to provide an oversimplified view on the DMD training dynamics, which are irrelevant to further introduction of the adaptive reward-based weighting.
3. I am not convinced that preference-based evaluation indicates that the proposed method is superior to the baselines, which didn’t use reward weighting during their training.
    1. The results presented in Table 2 do not seem to be convincing. 1) ImageReward and HPSv2 are known to be correlated and are hacked using higher CFG at the expense of over-saturation [1]. Could the authors demonstrate that this is not the case in their experiments? 2) Other methods do not utilise reward in any way. It would be unexpected of them to perform on par with the method that uses it for training.
    2. I believe DMD-R is the only baseline that uses reward during its training, but there is only one comparison with it on the GenEval benchmark, and it is unclear which of the methods gives better results. Therefore, a proper, extensive comparison with reward-based methods (e.g., DMD-R) is warranted.
4. Results in Table 4 don’t indicate a large improvement over the DMD baseline in terms of FID.
5. Minor: the teacher's distribution $p_{\text{real}}$ and student-induced $p_{\text{fake}}$ in Eq. 1, as well as their scores in Eq. 2, should depend on $t$.

## Significance / Originality
### Strengths
1. The reward-based weighting is novel in DMD distillation literature and represents a viable and promising algorithm, as indicated by the experiments.
### Weaknesses
1. The concept of *forbidden zones* is not researched properly. Therefore, there is little conceptual novelty, and the proposed method looks more like a heuristic than a formally justified algorithm.
## References
[1] Guidance Matters: Rethinking the Evaluation Pitfall for Text-to-Image Generation

---

> ### Author Rebuttal · Authors · 2026-03-31
>
> We sincerely thank the reviewer for the detailed and constructive feedback. Before addressing the specific concerns, we briefly restate the main contribution of AMD:
>
> Our intention is not to claim that the specific adaptive weighting rule in AMD is the unique formal consequence of the forbidden-zone analysis. Rather, our main goal is to provide **a perspective for understanding when and why vanilla DMD becomes unreliable**, and for subsuming existing methods into a unified view. Under this perspective, adaptive weighting is best seen as a practical instantiation of the principle: it enables simple and effective correction, rather than being the only closed-form solution implied by our work.
>
> ---
>
> **W1**: The connection between forbidden zones and adaptive weighting is unclear.
>
> **A**:  We thank the reviewer for pointing this out. Our intended logic is: (1) the forbidden-zone analysis explains why vanilla DMD can fail in a state-dependent way; (2) this suggests that samples should not all be treated identically; (3) AMD provides a simple practical mechanism for such adaptive correction.
>
> To further strengthen this connection, we also consider a reward-free variant which followes Definition 3.2, to detect whether a sample lies in the forbidden zone. AMD remains effective in this setting. Due to space limitations, details are deferred to our response to Reviewer SonN’s **W1**.
>
> ---
>
> **W2**: The Proposition 3.1 is oversimplified
>
> **A**: We thank the reviewer for this comment. The simplification is used as a first-order local approximation for trajectory analysis, $K_ \theta$ acts as a positive semi-definite preconditioning matrix, whose dominant local effect is to rescale the induced update while preserving the attractive–repulsive structure central to our analysis. Proposition 3.1 is therefore an interpretable local view, rather than a full characterization of DMD dynamics. We will make this scope and approximation more precise in the revised version.
>
> ---
>
> **W3**: Regarding the Table 2.
>
> - W3.a. HPSv2 and ImageReward are unreliable.
>
>   **A**: We agree that HPSv2 and ImageReward are imperfect. However, this particular issue is less direct in DMD-style distilled models, where **no extra CFG** is applied at inference time. More importantly, our conclusion is not based on these metrics alone: AMD also improves on rule-based metrics such as GenEval/VBench.
>
> - W3.b. Comparison results with DMDR are limited.
>
>   **A**: Beyond GenEval, we also compare against DMDR on the class-conditional generation task (Table 4). We would like to emphasize that our main focus is to analyze and unify DMD-style methods under a common mechanism-level view (with DMDR corresponding to the case illustrated in Fig. (b)), rather than to develop a general reward-optimization framework for diffusion models. We also note that works such as DMDR are similarly evaluated primarily against DMD-style baselines, which is consistent with common practice in this line of work.
>
> ---
>
> **W4**: The improvement in Table 4 is not significant.
>
> **A**: We agree that the improvement over DMD in Table 4 is not dramatic in absolute terms. However, further gains are inherently harder when the baseline FID is already very low. In this regime, AMD still consistently improves FID, sFID over DMD, indicating a stable gain. In contrast, DMDR achieves a much higher IS at the cost of substantially worse FID/sFID.
>
> ---
>
> **W5**: Typo Error
>
> **A**: We thank the reviewer for pointing this out. We will correct it in the revised version.
>
> ---
>
> **Q1**: Results in the unconditional setting
>
> **A**: We thank the reviewer for this question. The current AMD formulation is designed for conditional DMD-style distillation, since the CA/DM decomposition itself relies on conditional term. This is not unique to AMD: related methods such as D-DMD and DMD2 are also developed and validated in conditional settings. More importantly, the conditional term is not merely an auxiliary component, but one of the key factors enabling successful DMD-style distillation.
>
> ---
>
> **Q2**: How AMD affects the distillation dynamics at different noise levels?
>
> **A**:  As discussed in Proposition C.1, higher noise levels reduce the risk of samples falling into the Forbidden Zone. Motivated by this insight, we explored a simple AMD variant that **assigns higher noise levels to low-reward samples and lower noise levels to high-reward ones**. Preliminary results from the anonymous Fig.2 (<https://anonymous.4open.science/api/repo/anonymous-ICML-Submission-10939-CB92/file/index.html?v=7dabd7eb>) show that this variant leads to more stable training (lower gradient norms) and better performance (lower DMD loss), providing empirical support for Proposition C.1 within the AMD framework.
>
> ---
>
> We sincerely thank the reviewer for the constructive comments , which have helped us significantly improve the quality of this work. We look forward to the opportunity for further discussion.

---

> > ### Author Rebuttal · Reviewer_jg1g · 2026-04-03
> >
> > All of the raised concerns are addressed clearly. I appreciate the authors' clarification regarding the intended scope of the paper, the connection between forbidden zones and adaptive weighting, and experimental results in Table 2. However, I am still not entirely convinced by the theoretical grounding and significance of the unifying view on the DMD-based distillation methods that follows from Proposition 3.1. I appreciate that the authors will clarify the scope and approximations behind it in the revised version.
> > That said, I find that AMD is a useful and sufficiently well validated practical technique for distillation. Therefore, I am willing to raise my score to "Weak Accept".

---

> > > ### Author Response · Authors · 2026-04-04
> > >
> > > We sincerely thank you for your constructive comments.
> > >
> > > We are also glad to see that your all concerns are addressed clearly and you are willing to raise the score to "Weak Accept" explicitly.
> > >
> > > We would highly appreciate it if you manually raise the score accordingly consistent to the reply.

---

### Official Review · Reviewer_SonN · 2026-03-13

**Soundness:** 4
**Presentation:** 4
**Significance:** 4
**Originality:** 3
**Overall Recommendation:** 5
**Confidence:** 4

**Summary:**

This paper proposes a method to distill diffusion models for few-steps sampling.
It introduces a new method called Adaptive Matching Distillation (AMD), which rebalances attractive and repulsive forces in Distribution Matching Distillation (DMD).
First the authors identify Forbidden Zones, in which the DMD gradient either vanishes or is ill-defined.
Intuitively these zones corresponds out of distribution data for the teacher.
Furthermore, the paper introduces an interpretational framework that yields a clear taxonomy of recent DMD variants and simplifies the understanding of such variants and the design of further variants.
Finally, experimental results confirm their method performs very well, obtaining SotA on a majority of benchmarks.

**Compliance With Llm Reviewing Policy:**

Affirmed.

**Final Justification:**

I am keeping my score, my concerns have been addressed but they were minor and my score was already high.

**Key Questions For Authors:**

- Is the score evaluation a problem for large noises for the fake teacher, which unlike the reak teacher has not be smoothed with an EMA? Or is the fake teacher also smoothed by an EMA?
- Eq.13, is $s$ in $[0,1]$ or $\mathbb{R}^+$? $s>1$ makes $\beta$ negative turning the repulsive force into an attracting one.

**Limitations:**

Yes

**Strengths And Weaknesses:**

Strengths
- The interpretational framework is beautifully synthetic and powerful at expressing DMD variants clearly.
- The paper is very well written, with a focus on clarity.
- The method looks simple yet sophisticated, the apparent simplicity possibly being a result of the excellent presentation.
- The experimental results look very solid with lot of pictures to help ground the intuition behind the method.

Weaknesses
- The method has to rely on an external pre-trained reward neural network $R$, this feels unsatisfying even though this is not the only method to demand such a requirement.

---

> ### Author Rebuttal · Authors · 2026-03-31
>
> We sincerely thank the reviewer for the very positive assessment of our work and for recognizing the clarity of the paper, the unified interpretational framework, and the strength of the experimental results.  Below, we provide detailed explanations to address your concerns.
>
>
> **W1**: AMD incorporates an auxiliary pretrained reward model $R$.
>
> **A**:  We agree that introducing an $R$ adds an extra practical component. Our use of  $R$  is mainly engineering-motivated: although $R(x)$  does not perfectly align with the true data support, it provides an effective performance-efficiency trade-off for identifying likely forbidden-zone samples.
>
> As the reviewer also recognized, the AMD framework is not inherently tied to any external reward model. For example, as shown in [1], $\log p(x)$ can be approximated by using an ELBO-style score as $-\mathbb{E}_{t,\epsilon} \left[ \|| \epsilon - \epsilon _\theta (x_t) \||^2 \right] + C ,$ which depends only on the teacher itself. Another reviewer also suggested that the difference between real and fake teacher could be used as a proxy. Using this internal density proxy, we further conducted distillation experiments on Imagenet without introducing any external reward model, and observed improvements, as shown below.
>
> | Method | FID | sFID | IS |
> |---|---:|---:|---:|
> | DMD | 3.5573 | 5.8499 | 314.42 |
> | AMD (external proxy) | **3.4690** | 5.7464 | **316.02** |
> | AMD (intrinsic proxy) | 3.6860 | **5.4793** | 315.04 |
>
> Here, we observe that the intrinsic proxy improves sFID and IS, while slightly worsening FID. This is a reasonable outcome: the intrinsic proxy tends to favor higher-density regions of the training distribution, thereby making training more stable, at the cost of some diversity. This is also consistent with prior observations on DMD-style methods.
>
> In particular, we further visualized this intrinsic reward in a 2D toy experiment (see **external link** <https://anonymous.4open.science/api/repo/anonymous-ICML-Submission-10939-CB92/file/index.html?v=7dabd7eb> ). The visualization shows that the intrinsic reward covers the high-probability support regions of the teacher quite well. We hope this provides further evidence that intrinsic forbidden-zone diagnostics are both feasible and meaningful.
>
> More broadly, we hope this highlights a key value of AMD: it provides a unified view of distribution matching methods, in which forbidden-zone detection is one possible instantiation rather than a framework-specific requirement. In this sense, AMD is general and not tied to any particular proxy.
>
> ---
>
> **Q1**: Is the fake teacher also smoothed by an EMA?
>
> **A**: Thank you for this important question. The ema setting follows the underlying training framework. Specifically, for the DMD2 [2] experiments, we follow the official default configuration, which does not use EMA for the fake teacher; for WAN [3], we follow its standard training setup, which does use EMA. We will clarify this implementation detail in the revised version.
>
> Moreover, we believe that at larger noise levels, the fake and real teacher can actually exhibit greater distributional overlap, resulting in fewer forbidden-zone samples, as suggested by Fig. 2(c) and Proposition C.1. The intuition is that heavier noise makes the teacher distribution less concentrated and spreads its probability mass over a broader support. In the extreme case of pure noise, both teachers approach the same shared noise prior.
>
> ---
>
> **Q2**:  Clarity in Eq. 13
>
> **A**: Thank you for pointing this out. In Eq. 13, $s$ is intended to be in $ \[ 0,1 \] $. Therefore, $\beta$ remains non-negative and the corresponding term preserves its repulsive role. We will clarify this in the revised version.
>
> ---
>
> We sincerely thank the reviewer again for the positive assessment and insightful feedback.  We also look forward to any further discussion.
>
> ---
>
> **Reference**
>
> [1] Li, A. C., Prabhudesai, M., Duggal, S., Brown, E., & Pathak, D. (2023). Your diffusion model is secretly a zero-shot classifier. ICCV 2023.
>
> [2] Yin, T., Gharbi, M., Park, T., Zhang, R., Shechtman, E., Durand, F., & Freeman, W. T. (2024). Improved distribution matching distillation for fast image synthesis. NeurIPS 2024
>
> [3] Wan, T., Wang, A., Ai, B., Wen, B., Mao, C., Xie, C. W., ... & Liu, Z. (2025). Wan: Open and advanced large-scale video generative models. arXiv preprint arXiv:2503.20314.

---

> > ### Author Rebuttal · Reviewer_SonN · 2026-04-01
> >
> > All points addressed clearly.

---

> > > ### Author Response · Authors · 2026-04-07
> > >
> > > Thank you for your feedback and for the constructive review process. We sincerely appreciate your time, consideration, and valuable insights. We are glad that our clarifications addressed your concerns, and we truly appreciate your recognition of our work.

---

### Decision · Program_Chairs · 2026-04-30

**Decision:**

Accept (regular)

**Comment:**

AMD introduces a unified optimization framework for DMD-style distillation, identifying "Forbidden Zones" where the real teacher provides unreliable guidance and the fake teacher exerts insufficient repulsion. Based on this analysis, AMD uses reward proxies to detect and escape these regions via adaptive gradient reweighting and Repulsive Landscape Sharpening. The method is validated across image and video generation (SDXL, SiT, Wan2.1) with consistent improvements over DMD baselines. All four reviewers converged positively after rebuttal (scores: 5, 5, 4, 4), with three marking concerns as fully resolved. Key strengths identified across reviews:

  1. Strong unifying perspective. The reinterpretation of prior DMD variants as implicit strategies to avoid forbidden zones provides a clear and powerful taxonomy (SonN: "beautifully synthetic and powerful"). This framework both clarifies existing methods and motivates the AMD design.
  2. Well-validated practical technique. Comprehensive experiments across multiple modalities, architectures, and benchmarks demonstrate consistent gains. The method is simple to implement yet effective, with clear ablations on key design choices.
  3. Generality of the framework. The authors demonstrated during rebuttal that AMD is not tied to any specific external reward model, an intrinsic proxy based on teacher-fake teacher disagreement also works, addressing concerns about reward dependence (iobW, SonN). Robustness to noisy proxy signals was also shown.

The main residual concern (jg1g) is that the theoretical grounding via Proposition 3.1 relies on simplifying assumptions. The authors acknowledged this and will clarify the scope of the approximation in the camera-ready. This is a reasonable limitation for a paper whose primary contribution is a practical  framework supported by strong empirical evidence. The authors should also explicitly discuss computational overhead and the reward hacking trade-off in the final version.